# ACTSAFE: ACTIVE EXPLORATION WITH SAFETY CONSTRAINTS FOR REINFORCEMENT LEARNING

**Yarden As, Bhavya Sukhija** *
ETH Zürich

**Lenart Treven**
ETH Zürich

**Carmelo Sferrazza**
UC Berkeley

**Stelian Coros**
ETH Zürich

**Andreas Krause**
ETH Zürich

## ABSTRACT

Reinforcement learning (RL) is ubiquitous in the development of modern AI systems. However, state-of-the-art RL agents require extensive, and potentially unsafe, interactions with their environments to learn effectively. These limitations confine RL agents to simulated environments, hindering their ability to learn directly in real-world settings. In this work, we present ACTSAFE, a novel model-based RL algorithm for safe and efficient exploration. ACTSAFE learns a well-calibrated probabilistic model of the system and plans optimistically w.r.t. the epistemic uncertainty about the unknown dynamics, while enforcing pessimism w.r.t. the safety constraints. Under regularity assumptions on the constraints and dynamics, we show that ACTSAFE guarantees safety during learning while also obtaining a near-optimal policy in finite time. In addition, we propose a practical variant of ACTSAFE that builds on latest model-based RL advancements and enables safe exploration even in high-dimensional settings such as visual control. We empirically show that ACTSAFE obtains state-of-the-art performance in difficult exploration tasks on standard safe deep RL benchmarks while ensuring safety during learning.

## 1 INTRODUCTION

Reinforcement learning (RL) is a powerful paradigm for sequential decision-making under uncertainty, with many applications in games (Mnih et al., 2015; Silver et al., 2016), recommender systems (Maystre et al., 2023), nuclear fusion control (Degrave et al., 2022), data-center cooling (Lazic et al., 2018) and robotics (Lee et al., 2020; Brohan et al., 2022; Cheng et al., 2024). Despite the notable progress, its application without any use of simulators remains largely limited. This is primarily because, in many cases, RL methods require massive amounts of data for learning while also being inherently unsafe during exploration.

In many real-world settings, environments are complex and rarely align exactly with the assumptions made in simulators. Learning directly in the real world allows RL systems to close the sim-to-real gap and continuously adapt to evolving environments and distribution shifts. However, to unlock these advantages, RL algorithms must be sample-efficient and ensure safety throughout the learning process to avoid costly failures or risks in high-stakes applications. For instance, agents learning driving policies in autonomous vehicles must prevent collisions with other cars or pedestrians, even when adapting to new driving environments. This challenge is known as *safe exploration*, where the agent's exploration is restricted by safety-critical, often unknown, *constraints that must be satisfied throughout the learning process*.

Several works study safe exploration and have demonstrated state-of-the-art performance in terms of both safety and sample efficiency for learning in the real world (Sui et al., 2015; Wischnewski et al., 2019; Berkenkamp et al., 2021; Cooper & Netoff, 2022; Sukhija et al., 2023; Widmer et al., 2023). These methods maintain a "safe set" of policies during learning, selecting policies from this set to safely explore and gradually expand it. Under common regularity assumptions about the constraints, these approaches guarantee safety throughout learning. However, explicitly maintaining and expanding a safe set, limits these methods to low-dimensional policies, such as PID controllers. This makes them difficult to scale to more complex tasks such as those considered in deep RL.

---

*Equal contribution. Correspondence to: yardas@ethz.ch

The goal of this work is to address this gap. To this end, we propose a scalable model-based RL algorithm – ACTSAFE – for efficient and safe exploration. Crucially, ACTSAFE learns an uncertainty-aware dynamics model, which it uses to implicitly define and expand the safe set of policies. We theoretically show that ACTSAFE ensures safety throughout learning and converges to a near-optimal policy within a finite number of episodes. Moreover, ACTSAFE is practical and integrates seamlessly with state-of-the-art dynamics modeling techniques, for instance Dreamer (Hafner et al., 2023), delivering strong empirical performance. Thus, ACTSAFE advances the frontier of safe RL methods, both in theory and practice. Our main contributions are summarized below.

**Contributions**

- We propose ACTSAFE, a novel model-based RL algorithm for safe exploration in continuous state-action spaces. ACTSAFE maintains a *pessimistic* set of safe policies and *optimistically* selects policies within this set that yield trajectories with the largest model epistemic uncertainty.
- We show that when the dynamics lie in a reproducing kernel Hilbert space (RKHS), ACTSAFE guarantees safe exploration. In addition, we provide a sample-complexity bound for ACTSAFE, illustrating that ACTSAFE obtains $\epsilon$-optimal policies in a finite number of episodes. To the best of our knowledge, we are the first to show *safety and finite sample complexity* for safe exploration in model-based RL with continuous state-action spaces.
- In our experiments, we demonstrate that ACTSAFE, when combined with a Gaussian process dynamics model, achieves efficient and safe exploration. Additionally, we show that ACTSAFE scales to high-dimensional environments of the SAFETY-GYM and RWRL benchmarks, excelling in challenging exploration tasks with visual control while also incurring significantly fewer constraint violations.

## 2 RELATED WORKS

**Constrained Markov decision processes (CMDP) for safe RL**  Safety in reinforcement learning can be modeled in various ways (García et al., 2015; Brunke et al., 2022). Constrained Markov decision processes (CMDPs) serve as a natural option for this purpose, as they can encode unsafe behaviors through constraints and enjoy many classical results from planning in MDPs (Altman, 1999). Learning and planning in CMDPs have been extensively explored in the RL community, both theoretically and in practice. Notably, the works of Efroni et al. (2020); Vaswani et al. (2022); Ding et al. (2022); Müller et al. (2024) derive sample complexity bounds for CMDPs in discrete state-action spaces, whereas Achiam et al. (2017); Tessler et al. (2018); Stooke et al. (2020); Xu et al. (2021); Liu et al. (2022); As et al. (2022); Sootla et al. (2022); Huang et al. (2024) develop deep RL algorithms for CMDPs in continuous state-action spaces. However, all the aforementioned works relax the requirement of safe exploration and thus do not ensure the safety during learning. This is in contrast to this work, where we tackle the hard problem of safe exploration.

**Provably safe exploration**  Turchetta et al. (2016); Wachi et al. (2018); Wachi & Sui (2020) focus on safe exploration in CMDPs with *discrete* state-action spaces and a constraint function that lies in an RKHS. Zheng & Ratliff (2020) study sample complexity for safe exploration in discrete CMDPs. For continuous state-action spaces, Berkenkamp et al. (2021) and extensions thereof (Baumann et al., 2021; Sukhija et al., 2023; Hübotter et al., 2024), leverage ideas from safe Bayesian optimization (Sui et al., 2015) to directly optimize over the policy parameters in a model-free manner. The proposed algorithms guarantee safe exploration and finite sample complexity for learning an $\epsilon$-optimal solution. When evaluated on real-world systems, these methods exhibit remarkable sample efficiency while also being safe during learning (Cooper & Netoff, 2022; Kirschner et al., 2019; Widmer et al., 2023). However, these approaches are limited to simple low-dimensional policies, e.g., PID controllers, and are hard to scale to policies with more than few parameters. In a similar spirit, Berkenkamp et al. (2017) propose a model-based RL algorithm for safe learning, where safety is modeled in terms of Lyapunov stability. Even though the method enjoys similar theoretical guarantees as Berkenkamp et al. (2021), it assumes access to a generative simulator and thus cannot be applied to traditional online RL settings. In contrast, Koller et al. (2018); Curi et al. (2022) propose more practical safe learning methods in combination with model-predictive control (MPC). While these methods guarantee safety during learning, they lack optimality guarantees and are computationally expensive to run in real-time.

A common aspect among most of the aforementioned methods is their use of an intrinsic objective, such as the model epistemic uncertainty, to guide and restrain exploration. Crucially, these methods

maintain a safe set of policies which they gradually expand during learning by sampling policies that yield the highest intrinsic reward. In this work, we build on this key insight to propose a model-based RL algorithm for online learning that enjoys the same kind of guarantees while also being applicable in real-world settings such as deep RL.

**Safe exploration with deep RL** A common approach to safe exploration is the use of safety filters (Dalal et al., 2018; Wabersich & Zeilinger, 2021; Curi et al., 2022), which modify the actions produced by an unsafe policy to meet safety constraints before they are executed on the real system. A key advantage of safety filters is that they can be easily added to any "off-the-shelf" unsafe RL algorithm. However, while safety is ensured, safety filters can lead to arbitrarily bad exploration and therefore lack guarantees for optimality. The works of Srinivasan et al. (2020); Thananjeyan et al. (2021) rely on learning safety critics that certificate state-actions as safe, either for policy optimization or during online data collection. These works provide strong empirical results, including demonstrations of safe policies on real hardware. In addition, following this approach, Bharadhwaj et al. (2020) upper bound the the probability of making infeasible policy updates. Another line of work, relies on guaranteeing feasibility of policy optimization algorithms. Notably, Chow et al. (2019) use Lyapunov functions to guarantee feasibility of policy gradients iterates and derive their analysis on discrete state-action spaces. Usmanova et al. (2024) propose Log-Barriers SGD (LBSGD), an optimization algorithm that ensures feasibility of all its iterates with barrier functions, showcasing its application in navigation tasks with image observations. More recently, As et al. (2024); Ni & Kamgarpour (2024) use LBSGD for safe learning with *greedy* policy gradients, i.e., without considering intrinsic rewards to expand the safe set of policies. Crucially, this form of greedy policy search may result in sub-optimal policies, as described in Section 4 and empirically shown in Section 5.

## 3 Problem Setting

We consider a discrete-time, episodic, constrained Markov decision process (CMDP), where the goal is to find a policy that not only maximizes the reward but also keeps the accumulated costs below a specified threshold, i.e., satisfies a safety constraint. This type of formulation is common in real-world scenarios, such as robot navigation. In this setting, the reward could represent the negative distance to a target destination, while the costs could represent penalties, such as a cost of 1 incurred for each collision with an obstacle. The CMDP formulation allows us to separate these two objectives, thus ensuring constraint satisfaction and safety, for an optimal policy. In this setup, we consider dynamical systems with additive noise and bounded running rewards $r$ and costs $c$

$$s_{t+1} = f^*(s_t, a_t) + w_t, (s_t, a_t) \in \mathcal{S} \times \mathcal{A}, \ s(0) = s_0$$
$$r(s, a) \in [0, R_{\max}] \qquad \text{(Running reward)} \qquad (1)$$
$$c(s, a) \in [0, C_{\max}] \qquad \text{(Running cost)}.$$

Here $s_t \in \mathcal{S} \subset \mathbb{R}^{d_s}$ is the state, $a_t \in \mathcal{A} \subset \mathbb{R}^{d_a}$ the control input, and $w_t \in \mathcal{W} \subseteq \mathbb{R}^{d_s}$ the process noise. The dynamics $f^*$ are unknown and without loss of generality, the reward $r$ and cost $c$ are assumed to be known.

**Task** In this work, we study the following constrained RL problem (Altman, 1999)

$$\max_{\pi \in \Pi} J_r(\pi, f^*) := \max_{\pi \in \Pi} \mathbb{E}_{s_0, \pi} \left[ \sum_{t=0}^{T-1} r(s_t, a_t) \right] \text{ s.t. } J_c(\pi, f^*) := \mathbb{E}_{s_0, \pi} \left[ \sum_{t=0}^{T-1} c(s_t, a_t) \right] \le d; \quad (2)$$
$$s_{t+1} = f^*(s_t, \pi_t(s_t)) + w_t.$$

We study the episodic setting, with episodes $n = 1, \ldots, N$. At the beginning of the episode $n$, we deploy a policy $\pi_n = (\pi_{n,0}, \pi_{n,1}, \ldots, \pi_{n,T-1})$, chosen from the policy space $\Pi$ for a horizon of $T$ on the system. Next, we obtain the trajectory $\tau_n = (s_{n,0}, \ldots, s_{n,T})$, which we add to a dataset of transitions $\mathcal{D}_n = \{(z_{n,i} = (s_{n,i}, \pi_{n,i}(s_{n,i})), y_{n,i} = s_{n,i+1})_{0 \le i < T}\}$ and use the collected data to learn a model of $f^*$.

## 4 ActSafe: Active Exploration with Safety Constraints

A key challenge in learning with safety constraints is ensuring that these constraints are not violated during exploration. In the following, we introduce an idealized version of ACTSAFE, which guarantees safe exploration for dynamical systems with Gaussian process dynamics [1]. Moreover, we also

---

[1] These guarantees can be extended to more general well-calibrated models as in Curi et al. (2020)

provide a bound on the sample complexity of ACTSAFE for learning an $\epsilon$-optimal policy. To the best of our knowledge, this is the first model-based safe RL algorithm for continuous state-action spaces that provides guarantees for both safety and sample complexity. In Section 4.3, we discuss a practical variant scaling to more complex domains. Our choice of a model-based approach is motivated by its superior empirical sample efficiency (Chua et al., 2018; As et al., 2022) as well as our theoretical analysis.

## 4.1 ASSUMPTIONS

Theoretically studying safe exploration without any assumptions on the underlying dynamical system is an ill-posed problem. In the following, we make some assumptions on the underlying problem that are common in the model-based RL (Curi et al., 2020; Kakade et al., 2020) and safe RL (Berkenkamp et al., 2021; Baumann et al., 2021) literature.

**Assumption 4.1** (Continuity of $\boldsymbol{f}^*$ and $\boldsymbol{\pi}$). The dynamics model $\boldsymbol{f}^*$ is $L_{\boldsymbol{f}}$–Lipschitz, the cost $c$ is $L_c$–Lipschitz, and all $\boldsymbol{\pi} \in \Pi$ are continuous.

**Assumption 4.2** (Process noise distribution). The process noise is i.i.d. Gaussian with variance $\sigma^2$, i.e., $\boldsymbol{w}_t \overset{i.i.d}{\sim} \mathcal{N}(\boldsymbol{0}, \sigma^2 \mathbb{I})$.

We focus on the setting where $\boldsymbol{w}$ is homoscedastic for simplicity. However, our framework can also be applied to the more general heteroscedastic and sub-Gaussian case (Sukhija et al., 2024; Hübotter et al., 2024).

**Assumption 4.3** (Initial safe seed). We have access to an initial nonempty safe set $\mathcal{S}_0$ of policies, i.e., $\forall \boldsymbol{\pi} \in \mathcal{S}_0 : J_c(\boldsymbol{\pi}) \leq d$ and $\mathcal{S}_0 \neq \emptyset$.

This assumption is crucial since without any prior knowledge about the system, ensuring safety is unrealistic. Therefore, $\mathcal{S}_0$ allows us to start the learning process by selecting policies from this set. In practice, this safe set could be obtained from a simulator or offline demonstration data.

In the following, we assume that at each step $n$ we learn a mean estimate $\boldsymbol{\mu}_n$ of $\boldsymbol{f}^*$ and can quantify our uncertainty $\boldsymbol{\sigma}_n$ over the estimate. This allows us to learn an uncertainty-aware model of $\boldsymbol{f}^*$, which is crucial for exploration and safety. More formally, we learn a well-calibrated statistical model of $\boldsymbol{f}^*$ as defined below.

**Definition 4.4** (Well-calibrated statistical model of $\boldsymbol{f}^*$, Rothfuss et al. (2023)). Let $\mathcal{Z} \overset{\text{def}}{=} \mathcal{S} \times \mathcal{A}$. An all-time well-calibrated statistical model of the function $\boldsymbol{f}^*$ is a sequence $\{\mathcal{Q}_n(\delta)\}_{n \geq 0}$, where

$$\mathcal{Q}_n(\delta) \overset{\text{def}}{=} \left\{ \boldsymbol{f} : \mathcal{Z} \to \mathbb{R}^{d_s} \mid \forall \boldsymbol{z} \in \mathcal{Z}, \forall j \in \{1, \ldots, d_s\} : |\mu_{n,j}(\boldsymbol{z}) - f_j(\boldsymbol{z})| \leq \beta_n(\delta)\sigma_{n,j}(\boldsymbol{z}) \right\},$$

if, with probability at least $1 - \delta$, we have $\boldsymbol{f}^* \in \bigcap_{n \geq 0} \mathcal{Q}_n(\delta)$. Here, $f_j$, $\mu_{n,j}$ and $\sigma_{n,j}$ denote the $j$-th element in the vector-valued functions $\boldsymbol{f}$, $\boldsymbol{\mu}_n$ and $\boldsymbol{\sigma}_n$ respectively, and $\beta_n(\delta) \in \mathbb{R}_{\geq 0}$ is sequence of scalar functions that depends on the confidence level $\delta \in (0, 1]$ and is monotonically increasing in $n$.

Next, we assume that $\boldsymbol{f}^*$ resides in a Reproducing Kernel Hilbert Space (RKHS) of vector-valued functions and show that this is sufficient for us to obtain a well-calibrated model.

**Assumption 4.5.** We assume that the functions $f_j^*$, $j = \{1, \ldots, d_s\}$ lie in a RKHS with kernel $k$ and have a bounded norm $B$, that is $\boldsymbol{f}^* \in \mathcal{H}_{k,B}^{d_s}$, with $\mathcal{H}_{k,B}^{d_s} = \{\boldsymbol{f} \mid \|f_j\|_k \leq B, j = \{1, \ldots, d_s\}\}$. Moreover, we assume that $k(\boldsymbol{z}, \boldsymbol{z}) \leq \sigma_{\max}$ for all $\boldsymbol{z} \in \mathcal{Z}$.

Assumption 4.5 allows us to model $\boldsymbol{f}^*$ with GPs for which the mean and epistemic uncertainty $(\boldsymbol{\mu}_n(\boldsymbol{z}) = [\mu_{n,j}(\boldsymbol{z})]_{j \leq d_s}$, and $\boldsymbol{\sigma}_n(\boldsymbol{z}) = [\sigma_{n,j}(\boldsymbol{z})]_{j \leq d_s})$ have an analytical formula (c.f., Equation (9) in Appendix A).

**Lemma 4.6** (Well calibrated confidence intervals for RKHS, Rothfuss et al. (2023)). *Let $\boldsymbol{f}^* \in \mathcal{H}_{k,B}^{d_s}$. Suppose $\boldsymbol{\mu}_n$ and $\boldsymbol{\sigma}_n$ are the posterior mean and variance of a GP with kernel $k$ after episode $n$. There exists $\beta_n(\delta)$, for which the sequence $(\boldsymbol{\mu}_n, \boldsymbol{\sigma}_n, \beta_n(\delta))_{n \geq 0}$ represents a well-calibrated statistical model of $\boldsymbol{f}^*$.*

In summary, Assumption 4.5 and Lemma 4.6 show that in the RKHS setting, a GP is a well-calibrated model. For more general models like Bayesian neural networks (BNNs), methods such as Kuleshov et al. (2018) can be used for calibration. Overall, our results can also be extended beyond the RKHS setting to other classes of well-calibrated models similar to Curi et al. (2020).

## 4.2 ACTSAFE: ALGORITHMIC FRAMEWORK

A crucial element of safe exploration algorithms is the exploration–expansion dilemma (Hübotter et al., 2024). In the following, we explain this in further detail, we then present a sketch of ACTSAFE and finally the formal algorithm.

---

**Algorithm 1** ACTSAFE: ACTIVE EXPLORATION WITH SAFETY CONSTRAINTS (Expansion stage)

---

**Init:** Aleatoric uncertainty $\sigma$, Probability $\delta$, Statistical model $(\boldsymbol{\mu}_0, \boldsymbol{\sigma}_0, \beta_0(\delta))$
**for** episode $n = 1, \ldots, n^*$ **do**
$\quad \boldsymbol{\pi}_n = \arg\max_{\boldsymbol{\pi} \in \mathcal{S}_n} \max_{\boldsymbol{f} \in \mathcal{M}_n} \mathbb{E}_{\boldsymbol{\tau}^{\boldsymbol{\pi}, \boldsymbol{f}}} \left[ \sum_{t=0}^{T-1} \|\boldsymbol{\sigma}_{n-1}(\hat{\boldsymbol{s}}_t, \boldsymbol{\pi}(\hat{\boldsymbol{s}}_t))\| \right]$ ➤ Prepare policy
$\quad \mathcal{D}_n \leftarrow \text{ROLLOUT}(\boldsymbol{\pi}_n)$ ➤ Collect data
$\quad \text{Update } (\mathcal{M}_n, \mathcal{S}_n) \leftarrow \mathcal{D}_{1:n}$ ➤ Update statistical model and safe set
**end for**

---

**Safe set expansion**  To ensure the safety of the agent during the initial phases of learning, ACTSAFE begins exploration by selecting policies from $\mathcal{S}_0$ (Assumption 4.3). This reduces uncertainty $\boldsymbol{\sigma}_n$ about $\boldsymbol{f}^*$ for policies in $\mathcal{S}_0$, allowing us to infer the safety of policies beyond $\mathcal{S}_0$ and *expand* the safe set (see Figure 1). Safe set expansion is critical in safe RL because the optimal policy may lie outside the initial safe set, and expanding the safe set is necessary to reach it. Unlike traditional RL, where exploration focuses on maximizing reward, safe RL methods must also explore to expand the safe set. Methods like optimism and Thompson sampling, which focus on reward maximization, do not address this need for safe set expansion (Sui et al., 2015).

**Algorithm Sketch**  ACTSAFE operates in two stages; **(i)** expansion by intrinsic exploration and **(ii)** exploitation of extrinsic reward. In the first stage, ACTSAFE uses the model epistemic uncertainty as an intrinsic reward $r^{\text{explore}}(\boldsymbol{s}, \boldsymbol{a}) = \|\boldsymbol{\sigma}_{n-1}(\boldsymbol{s}, \boldsymbol{a})\|$ and selects policies within the safe set that yield trajectories with high uncertainties. This enables ACTSAFE to efficiently reduce its uncertainty within the safe set and expand it. ACTSAFE performs the intrinsic exploration phase for a fixed number of episodes $n^*$ till the safe set is sufficiently large and then transitions to the second stage. In the exploitation stage, ACTSAFE greedily maximizes the extrinsic reward $r$, effectively aiming to solve the problem in Equation (2).

Most model-based safe RL methods (As et al., 2022) focus only on the second stage and ignore safe set expansion. In contrast, the theoretically grounded approaches of Berkenkamp et al. (2021); Baumann et al. (2021); Sukhija et al. (2023); Hübotter et al. (2024) explicitly account for the expansion, but are not scalable to high dimensional policies typically considered in RL.

Next, we present our main algorithm. To ensure safety during learning, we maintain a conservative (pessimistic) estimate of the safe set which is defined below.

**Definition 4.7.** Let $\mathcal{M}_n \stackrel{\text{def}}{=} \mathcal{M}_{n-1} \cap \mathcal{Q}_n, \forall n \geq 1$ denote the set of plausible models, and $P_n(\boldsymbol{\pi}) = \max_{\boldsymbol{f} \in \mathcal{M}_n} J_c(\boldsymbol{\pi}, \boldsymbol{f})$ our *pessimistic* estimate of the expected costs w.r.t. $\mathcal{M}_n$. Then, we define the safe set $\mathcal{S}_n$ as

$$\mathcal{S}_n \stackrel{\text{def}}{=} \mathcal{S}_{n-1} \cup \{\boldsymbol{\pi} \in \Pi \setminus \mathcal{S}_{n-1}; \exists \boldsymbol{\pi}' \in \mathcal{S}_{n-1} \text{ s.t. } P_n(\boldsymbol{\pi}') + D(\boldsymbol{\pi}, \boldsymbol{\pi}') \leq d\}, \tag{3}$$

where

$$D(\boldsymbol{\pi}, \boldsymbol{\pi}') =$$
$$\mathbb{E}_{\boldsymbol{\tau}^{\boldsymbol{\pi}'}} \left[ \sum_{t=0}^{T-1} \min\left\{ L_c \|\boldsymbol{\pi}'(\boldsymbol{s}_t) - \boldsymbol{\pi}(\boldsymbol{s}_t)\|, 2C_{\max} \right\} + TC_{\max} \min\left\{ \frac{L_f \|\boldsymbol{\pi}'(\boldsymbol{s}_t) - \boldsymbol{\pi}(\boldsymbol{s}_t)\|}{\sigma}, 1 \right\} \right]$$

**Interpretation of Definition 4.7**  We maintain a pessimistic estimate, $P_n$ of the constraint value function $J_c$ w.r.t. our model set $\mathcal{M}_n$. In Equation (3) we define the expansion operator for the safe set. This operator adds new policies $\boldsymbol{\pi}$ that are not yet in the safe set, i.e., those in $\Pi \setminus \mathcal{S}_{n-1}$, to $\mathcal{S}_n$ if they are close to some policy $\boldsymbol{\pi}'$ from within the safe set. The distance $D(\boldsymbol{\pi}, \boldsymbol{\pi}')$ measures how close the two policies are in terms of the underlying cost function, and it is similar to other distance metrics, such as the one in Foster et al. (2024, Theorem 2.1).

The expansion operator is common in the safe BO and RL literature (Wischnewski et al., 2019; Fiducioso et al., 2019; Berkenkamp et al., 2021; Baumann et al., 2021; Cooper & Netoff, 2022; Sukhija et al., 2023; Holzapfel et al., 2024; Fiedler et al., 2024), and while it is generally difficult to evaluate in continuous spaces, it gives a key insight for safe RL methods: *to effectively expand our knowledge of what is safe, we have to reduce our pessimism across policies in our safe set.*

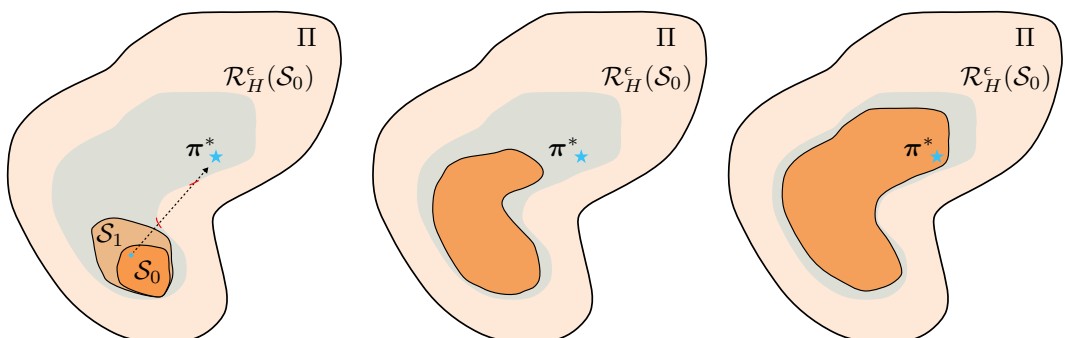

Figure 1: Schematic illustration of the expansion process. We expand the safe set at iteration $n-1$ by reducing our uncertainty around policies at the boundary of $\mathcal{S}_{n-1}$. The pale blue area depicts the reachable set $\mathcal{R}_H^\varepsilon(\mathcal{S}_0)$ after $H$ expansion iterations. The arrow on the leftmost illustration demonstrates that without explicit expansion, finding the optimal policy $\pi^*$ is intractable.

Accordingly, during the expansion phase, we use the following objective for ACTSAFE, which, in the $n$-th episode, selects the policy $\pi_n$ that yields the high uncertainty about the underlying dynamics

$$\pi_n, f_n = \underset{\pi \in \mathcal{S}_n, f \in \mathcal{M}_n}{\arg\max} \underbrace{\mathbb{E}_{\tau^{\pi,f}}\left[\sum_{t=0}^{T-1} \|\sigma_{n-1}(\hat{s}_t, \pi(\hat{s}_t))\|\right]}_{\overset{\text{def}}{=} J_{r_n}(\pi,f)}. \tag{4}$$

Furthermore, akin to Curi et al. (2020); Sukhija et al. (2024), we introduce additional exploration, by also optimistically picking the dynamics $f_n$ from our set of plausible models $\mathcal{M}_n$. Moreover, since the true dynamics, $f^*$ are unknown, we have to plan w.r.t. some dynamics model in $\mathcal{M}_n$. A theoretically grounded and well-established strategy for model-based RL methods is to pick an optimistic model $f_n$ from $\mathcal{M}_n$. As we show in Theorem 4.8 this results in first-of-its-kind sample complexity and safety guarantees. The expansion phase of the algorithm is summarized in Algorithm 1.

**Theorem 4.8.** *Let Assumptions 4.1 to 4.3 and 4.5 hold. Then, we have with probability at least $1-\delta$ that $J_c(\pi_n, f^*) \leq d \ \forall n \geq 0$, i.e., ACTSAFE is safe during all episodes.*

*Moreover, consider any $\epsilon > 0$ and define $\mathcal{R}_H^\epsilon(\mathcal{S}_0)$ as the reachable safe set after $H$ expansions*

$$\mathcal{R}_H^\epsilon(\mathcal{S}_0) \overset{\text{def}}{=} \mathcal{R}_{H-1}^\epsilon(\mathcal{S}_0) \cup \left\{\pi \in \Pi \setminus \mathcal{R}_{H-1}^\epsilon(\mathcal{S}_0); \exists \pi' \in \mathcal{R}_{H-1}^\epsilon(\mathcal{S}_0) \text{ s.t. } J_c(\pi') + D(\pi, \pi') \leq d - \epsilon\right\}$$
$$\mathcal{R}_0^\epsilon(\mathcal{S}_0) \overset{\text{def}}{=} \mathcal{S}_0.$$

*Let $n^*$ be the smallest integer such that*

$$\frac{n^*}{\gamma_{n^*}(k)\beta_{n^*}^4(\delta)} \geq \frac{(H+1)T^6 C^4 \frac{d_s \sigma_0^2}{\log(1+\sigma^{-2}\sigma_0^2)}}{\epsilon^2}., \tag{5}$$

*where $C = (1+\sqrt{d_s})\max\{C_{\max}, R_{\max}, \sigma_0\}$, $\gamma_n(k)$ the maximum information gain (Srinivas et al., 2012), and $\tilde{\pi}_n$ the solution to $\underset{\pi \in \mathcal{S}_n}{\arg\max} \min_{f \in \mathcal{M}_n} J_r(\pi, f)$. Then we have $\forall n \geq n^*$ with probability at least $1-\delta$*

$$\max_{\pi \in \mathcal{R}_H^\epsilon(\mathcal{S}_0)} J_r(\pi) - J_r(\tilde{\pi}_n) \leq \epsilon.$$

The theorem shows that ACTSAFE is safe during all episodes. Furthermore, it shows that after finishing the expansion phase, ACTSAFE achieves an $\epsilon$-optimal solution within $\mathcal{R}_H^\epsilon(\mathcal{S}_0)$ for the underlying reward function $r$, where $\mathcal{R}_H^\epsilon(\mathcal{S}_0)$ is the largest safe set we can obtain after $H$ expansion steps if we know the dynamics to $\epsilon$ precision. To the best of our knowledge, we are the first to prove safety and give sample complexity bounds for safe model-based RL algorithms in the episodic setting with continuous state-action spaces.

Intuitively, by maximizing the epistemic uncertainty, we explore our dynamics uniformly among all policies in the safe set $\mathcal{S}_n$, making our model more confident, i.e., reducing $\sigma_n$. As our uncertainty

---

**Algorithm 2 ACTSAFE: Practical Version**

---

**Init:** Model Set $\mathcal{Q}_0$
**for** episode $n = 1, \ldots, n^*$ **do**                                ➤ Intrinsic exploration phase
  Select $\boldsymbol{\pi}_n$ by solving Equation (7)                      ➤ Prepare policy
  $\mathcal{D}_n \leftarrow \text{ROLLOUT}(\boldsymbol{\pi}_n)$            ➤ Collect data
  Update $\mathcal{Q}_n \leftarrow \mathcal{D}_{1:n}$                      ➤ Update dynamics
**end for**
**for** episode $n = n^*, \ldots, N$ **do**                               ➤ Extrinsic exploration phase
  Select $\boldsymbol{\pi}_n$ by solving Equation (8)
  $\mathcal{D}_n \leftarrow \text{ROLLOUT}(\boldsymbol{\pi}_n)$
  Update $\mathcal{Q}_n \leftarrow \mathcal{D}_{1:n}$
**end for**

---

within $\mathcal{S}_n$ shrinks, we add more policies to our safe set (c.f. Definition 4.7) and thus facilitate its expansion. The proof of Theorem 4.8 is given in Appendix A.

While the algorithm itself is difficult to implement for continuous state-action spaces, it gives key insights that guide our practical implementation in Section 4.3: **(i)** maximization of intrinsic rewards for expansion, **(ii)** pessimism w.r.t. plausible dynamics to define a safe set of policies $\mathcal{S}_n$, and **(iii)** selecting $\boldsymbol{\pi}_n$ only from $\mathcal{S}_n$ to ensure safety. Building on these insights, we introduce a practical version of ACTSAFE designed to excel in real-world scenarios, such as visual control tasks.

### 4.3 PRACTICAL IMPLEMENTATION

**Optimizing over safe policies** In Equation (4) we optimize over the policies within the safe set, where the safe set is defined according to Definition 4.7. This is particularly challenging in continuous state-action spaces since it requires us to maintain the model set $\mathcal{M}_n$ and the safe set $\mathcal{S}_n$. We modify the definition of the safe set which makes the optimization problem more tractable.

$$\widehat{\mathcal{S}}_n = \left\{ \boldsymbol{\pi} \in \Pi; \text{ s.t. } \max_{\boldsymbol{f}' \in \mathcal{Q}_n} J_c(\boldsymbol{\pi}, \boldsymbol{f}') \leq d \right\} \tag{6}$$

Note that $\widehat{\mathcal{S}}_n \subseteq \mathcal{S}_n$, making it a conservative estimate of $\mathcal{S}_n$, therefore selecting policies from $\widehat{\mathcal{S}}_n$ still preserves the safety guarantees. Furthermore, in $\widehat{\mathcal{S}}_n$, we are pessimistic w.r.t. the dynamics $\boldsymbol{f} \in \mathcal{Q}_n$ and thus we can simply use $\boldsymbol{\mu}_n, \boldsymbol{\sigma}_n$ to induce pessimism, i.e., we do not have to maintain the model set $\mathcal{M}_n = \mathcal{M}_{n-1} \cap \mathcal{Q}_n$ (c.f. Definition 4.4). A similar relaxation is made by other safe RL algorithms such as Berkenkamp et al. (2021); Baumann et al. (2021).

To practically solve Equation (4) we use $\widehat{\mathcal{S}}_n$ instead of $\mathcal{S}_n$, resulting in the following problem

$$\arg\max_{\boldsymbol{\pi} \in \Pi} \max_{\boldsymbol{f} \in \mathcal{Q}_n} J_n(\boldsymbol{\pi}, \boldsymbol{f}) \text{ s.t. } \max_{\boldsymbol{f}' \in \mathcal{Q}_n} J_c(\boldsymbol{\pi}, \boldsymbol{f}') \leq d. \tag{7}$$

Equation (7) is a constrained optimization problem with the added complexity of optimizing over the dynamics in $\mathcal{Q}_n$. Moreover, it does not require us to maintain $\widehat{\mathcal{S}}_n$ since we implicitly account for it in the constraint in Equation (7), making it tractable for continuous state-action spaces. In Equation (7), we have to solve $\max_{\boldsymbol{f}' \in \mathcal{Q}_n} J_c(\boldsymbol{\pi}, \boldsymbol{f}')$ to enforce pessimism for safety. To this end, we use the methods from Yu et al. (2020) for our experiments. In practice, we solve Equation (7) by using a CMDP planner based on Log-Barrier SGD (LBSGD, Usmanova et al., 2024). Further technical details can be found in Appendix B.

**From CMDPs to visual control** ACTSAFE can be seamlessly integrated with state-of-the-art model-based RL methods for learning in visual control tasks (Hafner et al., 2019; 2023). To tighten the gap between RL and real-world problems, we relax the typical full observability assumption and consider problems where the agent receives an observation $\boldsymbol{o}_t \sim p(\cdot | \boldsymbol{s}_t)$ instead of $\boldsymbol{s}_t$ at each time step. To handle partial observability, we choose to base our dynamics model on the Recurrent State Space Model (RSSM) introduced in Hafner et al. (2019). The RSSM can be thought of as a sequential variational auto-encoder that learns the (latent) dynamics $\boldsymbol{f}$. We leverage approximate Bayesian inference techniques, in particular probabilistic ensembles (Lakshminarayanan et al., 2017), to approximate the posterior $p(\boldsymbol{f} | \mathcal{D}_n)$ over RSSMs. In particular, we learn an ensemble of $M$ models and define $\mathcal{Q}_n$ as $\mathcal{Q}_n = \bigcup_{i=0}^{M-1} \{\boldsymbol{f}^i\}$. The model's epistemic uncertainty (disagreement) is then used to enforce pessimism w.r.t. the safety constraints and for the intrinsic reward exploration (see Algorithm 2).

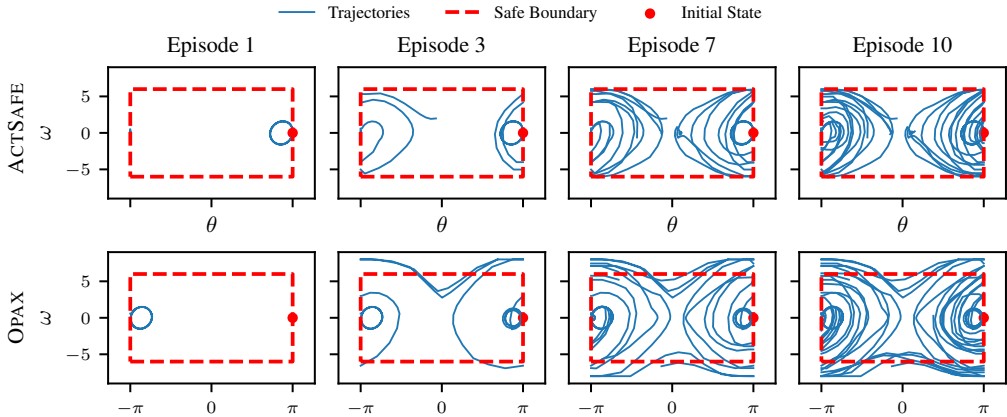

Figure 2: Safe exploration in the PENDULUMSWINGUP task. Each plot above visualizes trajectories considered during exploration across all past learning episodes. The red box in the plot depicts the safety boundary in the state space. ACTSAFE maintains safety throughout learning.

**Online policy improvement**   After the first intrinsic exploration phase, it is often necessary to perform additional learning updates during the second exploitation phase (Sekar et al., 2020). Therefore, after $n^*$ iterations of intrinsic exploration, we optimize the extrinsic reward by solving

$$\arg\max_{\boldsymbol{\pi}\in\Pi} \max_{\boldsymbol{f}\in\mathcal{Q}_n} J_r(\boldsymbol{\pi}, \boldsymbol{f}) \text{ s.t. } \max_{\boldsymbol{f}'\in\mathcal{Q}_n} J_c(\boldsymbol{\pi}, \boldsymbol{f}') \leq d. \tag{8}$$

## 5 EXPERIMENTS

In the following, we evaluate ACTSAFE on state-based and visual control tasks. For the state-based tasks, we use GPs to model the dynamics $\boldsymbol{f}^*$. For the visual control tasks, we use the RSSM model from Hafner et al. (2019) as described in Section 4.3. We thus validate both the theoretical and practical aspects of ACTSAFE in this section.

### 5.1 DOES ACTSAFE EXPLORE SAFELY WITH GPS?

We evaluate ACTSAFE on the PENDULUM and CARTPOLE environments. Additional details on the experimental setup, including the safety constraints, are provided in Appendix B. For both environments, we run the algorithms for ten episodes and then use the learned model to plan w.r.t. known extrinsic rewards after the expansion phase. For extrinsic rewards, we consider the SWINGUP task. We study the effects of pessimism with respect to the model uncertainty for safety. To this end, we consider as baselines a version of ACTSAFE without pessimism, which only uses the mean model $\boldsymbol{\mu}_n$ for planning and OPAX (Sukhija et al., 2024), an unsafe active exploration algorithm.

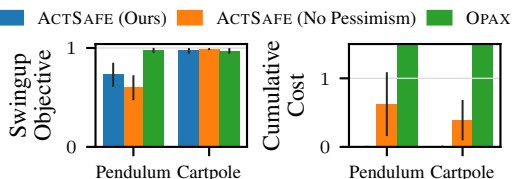

Figure 3: Evaluation of safety via pessimism and intrinsic exploration. The cumulative cost accumulates all the incurred costs during learning, the reported objective performance is normalized. ACTSAFE maintains safety during learning while attaining high zero-shot performance on the PENDULUMSWINGUP objective at test time.

We present our results in Figure 3, where we report the performance and the total accumulated costs during exploration of our method. We conclude that ACTSAFE does not incur any costs during learning. In contrast, the variant of ACTSAFE without pessimism and OPAX are unsafe during learning. This validates the necessity of using the model epistemic uncertainty to enforce pessimism during exploration. Note that ACTSAFE pays a price in terms of performance for pessimism, as it converges to a lower reward value than the other algorithms.

In Figure 2 we visualize the trajectories of ACTSAFE and OPAX in the state space during exploration. We observe that both algorithms cover the state space well, however, ACTSAFE remains within the safety boundary during learning whereas OPAX violates the constraints.

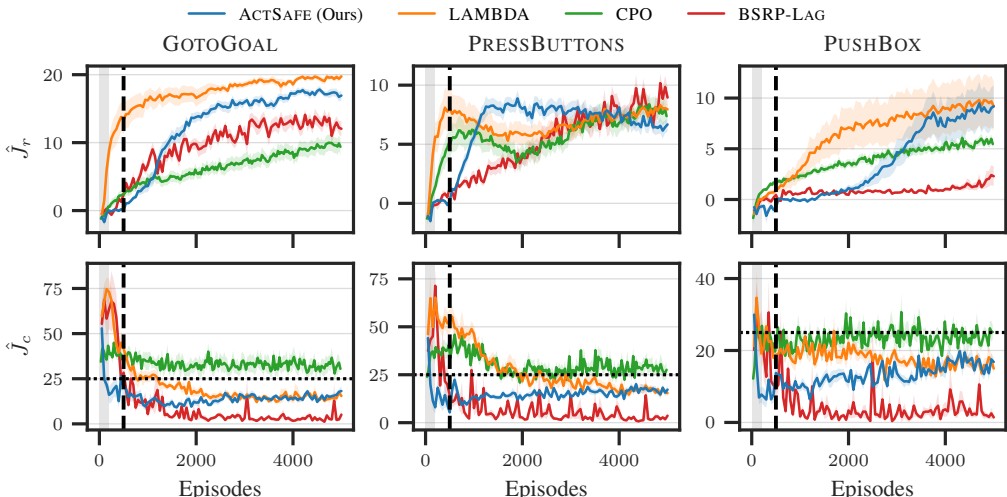

Figure 4: Safety of ACTSAFE in SAFETY-GYM with vision control. The dotted horizontal line depicts the safety constraint. We report the mean and standard error across 10 seeds. The vertical dashed line illustrates the transition of ACTSAFE from the intrinsic exploraiton/expansion phase to the extrinsic reward phase. Grey shaded area represents the warm-up phase.

## 5.2 DOES ACTSAFE SCALE TO VISION CONTROL?

While with GPs we can work closer to theory, scaling them to high dimensions with large data regimes, in particular visual control tasks, is challenging. We demonstrate our practical implementation (Algorithm 2) on high-dimensional RL tasks. We highlight that ensuring safety with an NN model with randomly initialized weights is impractical without any additional prior knowledge. To this end, for all experiments hereon, we assume access to an initial data collection (warm-up) period of 200K environment steps, where the agent collects data and uses it to calibrate its world model. This experimental setup is simple as it seamlessly integrates with both off and on-policy algorithms, such as CPO. Furthermore, it simulates a realistic setting, where the agent can collect some data initially in a controlled/supervised setting where safety is not directly penalized. However, after the initial data collection period, the agent is required to be safe during learning. We use the same training procedure across all baselines and environments (akin to Dalal et al., 2018). In Appendix C, we present additional experiments that study safe exploration under distribution shifts of the dynamics, effectively leveraging the simulator to calibrate the model and imitating sim-to-real transfer.

**Safety** We investigate ACTSAFE's performance in terms of constraint satisfaction during learning and compare it with state-of-the-art baseline algorithms for safe vision control (As et al., 2022; Huang et al., 2024) and with CPO (Achiam et al., 2017). We use the same experimental setup from SAFETY-GYM (Ray et al., 2019) and As et al. (2022), with the POINT robot in all tasks. As shown in Figure 4, compared to the baselines, ACTSAFE, significantly reduces constraint violation on all tasks. Notably, while ACTSAFE slightly underperforms LAMBDA, it incurs much smaller costs. This result may be interpreted by the conservatism needed to maintain safety during learning. Furthermore, we observe that BSRP-LAG generally underperforms both algorithms in terms of safety and performance. We provide more details on our comparison in Appendix B. Additionally, we ablate our choice of LBSGD in Appendix C and highlight its benefits.

**Exploration** In this experiment, we examine the influence of using an intrinsic reward in hard exploration tasks. To this end, we extend tasks from SAFETY-GYM and introduce three new tasks with sparse rewards, i.e., without any reward shaping to guide the agent to the goal. We provide more details about the rewards in Appendix B. In Figure 5 we compare ACTSAFE with a GREEDY baseline that collects trajectories only based on the sparse extrinsic reward. As shown, ACTSAFE substantially outperforms GREEDY in all tasks, while violating the constraint only once in the GOTOGOAL task. In addition to SAFETY-GYM, we evaluate on CARTPOLESWINGUPSPARSE from RWRL (Dulac-Arnold et al., 2019) with additional penalty for large actions (see Curi et al., 2020, and Appendix B). We compare ACTSAFE with three baselines. **(i)** UNIFORM, which samples actions uniformly at random during exploration, **(ii)** OPTIMISTIC, which uses the model epistemic uncertainty estimates as exploration reward bonuses and **(iii)** GREEDY, which optimizes the extrinsic reward directly.

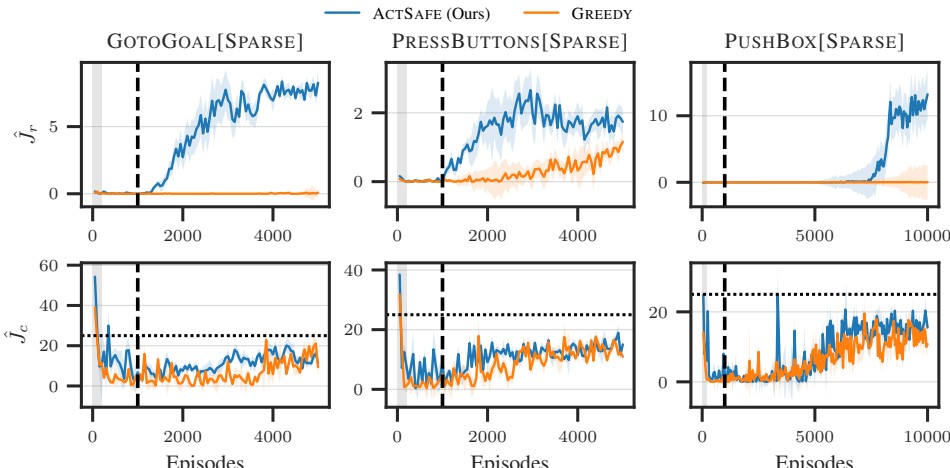

Figure 5: Performance on hard safe exploration tasks. The vertical dashed line illustrates the transition from data collection with intrinsic to extrinsic reward.

Figure 6 indicates that using uncertainty quantification for exploration is crucial, as only ACTSAFE and OPTIMISTIC find non-trivial policies. Despite that, ACTSAFE outperforms OPTIMISTIC. Furthermore, even though UNIFORM initially explores with an unsafe policy, it is insufficient to learn a good dynamics model, and thus underperforms ACTSAFE. This is mainly due to the undirected exploration strategy of UNIFORM, which does not leverage the model's epistemic uncertainty.

**Discussion** Our experiments underscore the following key findings. First, intuitively, in the GP setting, where our implementation is closer to theory, pessimism w.r.t. the model uncertainty plays a crucial role as we achieve strict safe exploration. Second, in our visual control experiments, using a small fraction of data ($<5\%$ of total data collected) as the warm-up period for calibrating the model and policy is sufficient for drastically reducing constraint violation. Learning safely typically requires some form of prior knowledge about

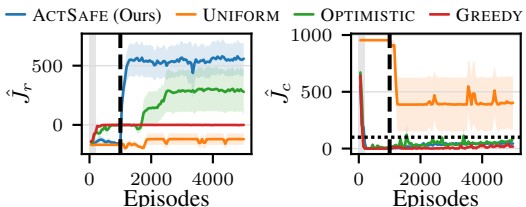

Figure 6: Hard exploration performance in CART-POLESWINGUPSPARSE of RWRL benchmark. We report the mean and standard error.

the problem, hence, using the data from the warm-up period keeps the experiment setup realistic without imposing specific domain knowledge and thus sacrificing generality. Third, in addition to exploring safely ACTSAFE, also solves tough exploration problems with the intrinsic rewards playing a crucial role. These results underline the importance of intrinsic exploration in RL, especially in safety-critical tasks. Moreover, ACTSAFE transfers directly from the GP setting to the vision control setting and in both cases our results show that ACTSAFE outperforms the baselines in terms of both safety and performance. We provide additional experiments in Appendix C, where ablate our choice of the LBSGD planner, evaluate ACTSAFE on a setting with distribution shifts in the dynamics and on a realistic robotics task from the state-of-the-art humanoid benchmark from Sferrazza et al. (2024).

## 6 CONCLUSIONS

In this paper, we introduce ACTSAFE, a safe model-based RL algorithm that leverages epistemic uncertainty as an intrinsic reward to learn a dynamics model efficiently and safely. We theoretically study systems with continuous state-action spaces and non-linear dynamics that lie in the RKHS, and provide guarantees on safety and near-optimality. We derive a practical variant of ACTSAFE, and demonstrate safe exploration and competitive performance with a Gaussian process dynamics model. Furthermore, we identify the key concepts that enable safe exploration with ACTSAFE and demonstrate how one can heuristically apply them to solve high-dimensional safe RL problems. Our empirical results showcase the importance of intrinsic rewards in the context of safety, demonstrating that ACTSAFE outperforms the baselines in the majority of tasks. In conclusion, ACTSAFE represents a significant advancement in safe reinforcement learning methods, enhancing both theoretical insights and practical applications.

ACKNOWLEDGMENTS

We thank Jonas Hübotter for the insightful discussion and feedback on this work. This project has received funding from the Swiss National Science Foundation under NCCR Automation, grant agreement 51NF40 180545, the Microsoft Swiss Joint Research Center, grant of the Hasler foundation (grant no. 21039) and the SNSF Postdoc Mobility Fellowship 211086.

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

## A   PROOFS

In the following, we prove Theorem 4.8. First, we provide the analytical formula for the mean and epistemic uncertainty of a GP model. We denote $\boldsymbol{x} := (\boldsymbol{s}, \boldsymbol{a})$, so that

$$
\begin{aligned}
\mu_{n,j}(\boldsymbol{x}) &= \boldsymbol{k}_n^\top(\boldsymbol{x})(\boldsymbol{K}_n + \sigma_\epsilon^2 \boldsymbol{I})^{-1} \boldsymbol{y}_{n,j} \\
\sigma_{n,j}^2(\boldsymbol{x}) &= k(\boldsymbol{x}, \boldsymbol{x}) - \boldsymbol{k}_n^T(\boldsymbol{x})(\boldsymbol{K}_n + \sigma_\epsilon^2 \boldsymbol{I})^{-1} \boldsymbol{k}_n(\boldsymbol{x})
\end{aligned}
\tag{9}
$$

where $\boldsymbol{y}_{n,j} = [s'_{i,j}]_{i \leq n}^\top$ is the vector of the $j$-th element of the observed next states $\boldsymbol{s}'_i$, $\boldsymbol{k}_n(\boldsymbol{x}) = [k(\boldsymbol{x}, \boldsymbol{x}_i)]_{i \leq n}^\top$, and $\boldsymbol{K}_n = [k(\boldsymbol{x}_i, \boldsymbol{x}_l)]_{i,l \leq n}$ is the kernel matrix. By concatenating the element-wise posterior mean and standard deviation, we obtain $\boldsymbol{\mu}_n(\boldsymbol{x}) = [\mu_{n,j}(\boldsymbol{x})]_{j \leq d_s}^\top$ and $\boldsymbol{\sigma}_n(\boldsymbol{x}) = [\sigma_{n,j}(\boldsymbol{x})]_{j \leq d_s}^\top$.

**Corollary A.1.** *Let assumption 4.5 hold, then we have for all $\boldsymbol{\pi} \in \Pi$, $n \geq 0$, with probability at least $1 - \delta$*

$$
P_n(\boldsymbol{\pi}) \geq J_c(\boldsymbol{\pi}, \boldsymbol{f}^*)
$$

*Proof.* Note that $\boldsymbol{f}^* \in \mathcal{Q}_n$ for all $n \geq 0$ with probability at least $1 - \delta$ (Lemma 4.6). Therefore, $\boldsymbol{f}^* \in \mathcal{M}_n$. Furthermore,

$$
\begin{aligned}
P_n(\boldsymbol{\pi}) &= \max_{\boldsymbol{f} \in \mathcal{M}_n} J_c(\boldsymbol{\pi}, \boldsymbol{f}) \\
&\geq J_c(\boldsymbol{\pi}, \boldsymbol{f}^*)
\end{aligned}
$$

$\square$

**Lemma A.2** (Difference in Policy performance, Sukhija et al. (2024)). *Consider any function $r : \mathcal{S} \times \mathcal{A} \to \mathbb{R}$. Let $J_{r,k}(\boldsymbol{\pi}, \boldsymbol{f}^*, \boldsymbol{s}_k) = \mathbb{E}_{\boldsymbol{\tau}^{\boldsymbol{\pi}}} \left[ \sum_{t=k}^{T-1} r(\boldsymbol{s}_t, \boldsymbol{\pi}(\boldsymbol{s}_t)) \right]$ and $A_{r,k}(\boldsymbol{\pi}, \boldsymbol{s}, \boldsymbol{a}) = \mathbb{E}_{\boldsymbol{\tau}^{\boldsymbol{\pi}}} [r(\boldsymbol{s}, \boldsymbol{a}) + J_{r,k+1}(\boldsymbol{\pi}, \boldsymbol{f}^*, \boldsymbol{s}') - J_{r,k}(\boldsymbol{\pi}, \boldsymbol{f}^*, \boldsymbol{s})]$ with $\boldsymbol{s}' = \boldsymbol{f}^*(\boldsymbol{s}, \boldsymbol{a}) + \boldsymbol{w}$. For simplicity we refer to $J_{r,0}(\boldsymbol{\pi}, \boldsymbol{f}^*, \boldsymbol{s}_0) = J_r(\boldsymbol{\pi}, \boldsymbol{f}^*, \boldsymbol{s}_0)$. The following holds for all $\boldsymbol{s}_0 \in \mathcal{S}$:*

$$
J_r(\boldsymbol{\pi}', \boldsymbol{f}^*, \boldsymbol{s}_0) - J_r(\boldsymbol{\pi}, \boldsymbol{f}^*, \boldsymbol{s}_0) = \mathbb{E}_{\boldsymbol{\tau}^{\boldsymbol{\pi}'}} \left[ \sum_{t=0}^{T-1} A_{r,t}(\boldsymbol{\pi}, \boldsymbol{s}'_t, \boldsymbol{\pi}'(\boldsymbol{s}'_t)) \right]
$$

*Proof.* See Lemma 5. Sukhija et al. (2024). $\square$

**Lemma A.3** (Comparing safety costs of policies).

$$
J_c(\boldsymbol{\pi}, \boldsymbol{f}^*, \boldsymbol{s}_0) - J_c(\boldsymbol{\pi}', \boldsymbol{f}^*, \boldsymbol{s}_0) \leq D(\boldsymbol{\pi}, \boldsymbol{\pi}')
$$

*Proof.* For notational convenience we will omit the dependance on $\boldsymbol{f}^*$.

$$J_c(\boldsymbol{\pi}, \boldsymbol{s}_0) - J_c(\boldsymbol{\pi}', \boldsymbol{s}_0) = \mathbb{E}_{\boldsymbol{\tau}\boldsymbol{\pi}'} \left[ \sum_{t=0}^{T-1} -A_{c,t}(\boldsymbol{\pi}, \boldsymbol{s}'_t, \boldsymbol{\pi}'(\boldsymbol{s}'_t)) \right]$$

$$= \mathbb{E}_{\boldsymbol{\tau}\boldsymbol{\pi}'} \left[ \sum_{t=0}^{T-1} -(c(\boldsymbol{s}_t, \boldsymbol{\pi}'(\boldsymbol{s}_t)) - c(\boldsymbol{s}_t, \boldsymbol{\pi}(\boldsymbol{s}_t))) \right]$$

$$+ \mathbb{E}_{\boldsymbol{\tau}\boldsymbol{\pi}'} \left[ \sum_{t=0}^{T-1} \mathbb{E}_{\boldsymbol{s}'_{t+1}|\boldsymbol{s}_t, \boldsymbol{\pi}(\boldsymbol{s}_t)} \left[ J_{c,k+1}(\boldsymbol{\pi}, \boldsymbol{s}'_{t+1}) \right] - \mathbb{E}_{\boldsymbol{s}'_{t+1}|\boldsymbol{s}_t, \boldsymbol{\pi}'(\boldsymbol{s}_t)} \left[ J_{c,k+1}(\boldsymbol{\pi}, \boldsymbol{s}'_{t+1}) \right] \right]$$

$$= \mathbb{E}_{\boldsymbol{\tau}\boldsymbol{\pi}'} \left[ \sum_{t=0}^{T-1} c(\boldsymbol{s}_t, \boldsymbol{\pi}(\boldsymbol{s}_t)) - c(\boldsymbol{s}_t, \boldsymbol{\pi}'(\boldsymbol{s}_t)) \right]$$

$$+ \mathbb{E}_{\boldsymbol{\tau}\boldsymbol{\pi}'} \left[ \sum_{t=0}^{T-1} \mathbb{E}_{\boldsymbol{s}'_{t+1}|\boldsymbol{s}_t, \boldsymbol{\pi}(\boldsymbol{s}_t)} \left[ J_{c,k+1}(\boldsymbol{\pi}, \boldsymbol{s}'_{t+1}) \right] - \mathbb{E}_{\boldsymbol{s}'_{t+1}|\boldsymbol{s}_t, \boldsymbol{\pi}'(\boldsymbol{s}_t)} \left[ J_{c,k+1}(\boldsymbol{\pi}, \boldsymbol{s}'_{t+1}) \right] \right]$$

$$\leq \mathbb{E}_{\boldsymbol{\tau}\boldsymbol{\pi}'} \left[ \sum_{t=0}^{T-1} \min \left\{ L_c \left\| \boldsymbol{\pi}'(\boldsymbol{s}_t) - \boldsymbol{\pi}(\boldsymbol{s}_t) \right\|, 2C_{\max} \right\} \right]$$

$$+ \mathbb{E}_{\boldsymbol{\tau}\boldsymbol{\pi}'} \left[ \sum_{t=0}^{T-1} \sqrt{\mathbb{E}_{\boldsymbol{s}'_{t+1}|\boldsymbol{s}_t, \boldsymbol{\pi}(\boldsymbol{s}_t)} \left[ J^2_{c,k+1}(\boldsymbol{\pi}, \boldsymbol{s}'_{t+1}) \right]} \min \left\{ \frac{\left\| \boldsymbol{f}^*(\boldsymbol{s}_t, \boldsymbol{\pi}(\boldsymbol{s}_t)) - \boldsymbol{f}^*(\boldsymbol{s}_t, \boldsymbol{\pi}'(\boldsymbol{s}_t)) \right\|}{\sigma}, 1 \right\} \right]$$

(Kakade et al., 2020, Lemma C.2.)

$$\leq \mathbb{E}_{\boldsymbol{\tau}\boldsymbol{\pi}'} \left[ \sum_{t=0}^{T-1} \min \left\{ L_c \left\| \boldsymbol{\pi}'(\boldsymbol{s}_t) - \boldsymbol{\pi}(\boldsymbol{s}_t) \right\|, 2C_{\max} \right\} + T C_{\max} \min \left\{ \frac{L_{\boldsymbol{f}} \left\| \boldsymbol{\pi}'(\boldsymbol{s}_t) - \boldsymbol{\pi}(\boldsymbol{s}_t) \right\|}{\sigma}, 1 \right\} \right]$$

$$= D(\boldsymbol{\pi}, \boldsymbol{\pi}')$$

□

**Lemma A.4.** *Let assumption 4.1 – assumption 4.5 hold. Then we have $\forall n \geq 0$, $\boldsymbol{\pi} \in \mathcal{S}_n \setminus \mathcal{S}_{n-1}$ with probability at least $1 - \delta$, $J_c(\boldsymbol{\pi}) \leq d$.*

*Proof.* Consider any $\boldsymbol{\pi} \in \mathcal{S}_n \setminus \mathcal{S}_{n-1}$. By Definition 4.7, we have that there exists a $\boldsymbol{\pi}'$ in $\mathcal{S}_{n-1}$ such that

$$P_n(\boldsymbol{\pi}') + D(\boldsymbol{\pi}, \boldsymbol{\pi}') \leq d$$

Therefore,

$$\begin{aligned} d &\geq P_n(\boldsymbol{\pi}') + D(\boldsymbol{\pi}, \boldsymbol{\pi}') \\ &\geq J_c(\boldsymbol{\pi}', \boldsymbol{f}^*) + D(\boldsymbol{\pi}, \boldsymbol{\pi}') &&\text{(Corollary A.1)} \\ &\geq J_c(\boldsymbol{\pi}, \boldsymbol{f}^*). &&\text{(Lemma A.3)} \end{aligned}$$

□

**Corollary A.5** (All policies in $\mathcal{S}_n$ are safe)**.** *Let assumption 4.1 – assumption 4.5 hold. Then we have $\forall n \geq 0$, $\boldsymbol{\pi} \in \mathcal{S}_n$ with probability at least $1 - \delta$, $J_c(\boldsymbol{\pi}) \leq d$.*

*Proof.* We prove this by induction. For $n = 0$, this holds due to assumption 4.3. Consider any $n > 0$. By induction, $\forall \boldsymbol{\pi} \in \mathcal{S}_n$ we have that $J_c(\boldsymbol{\pi}, \boldsymbol{f}^*) \leq d$. Hence, we focus on $\boldsymbol{\pi} \in \mathcal{S}_{n+1} \setminus \mathcal{S}_n$. In Lemma A.4, we show $J_c(\boldsymbol{\pi}, \boldsymbol{f}^*) \leq d$ for all $\boldsymbol{\pi} \in \mathcal{S}_{n+1} \setminus \mathcal{S}_n$. This completes the proof. □

**Lemma A.6.** *Consider any positive and bounded function $c \in [0, C_{\max}]$. Let assumption 4.1 – 4.5 hold. Then we have $\forall n \geq 0$, $\forall \boldsymbol{f} \in \mathcal{M}_n$. with probability at least $1 - \delta$*

$$|J_c(\boldsymbol{\pi}, \boldsymbol{f}) - J_c(\boldsymbol{\pi}, \boldsymbol{f}^*)| \leq T C_{\max} \mathbb{E}_{\boldsymbol{\tau}\boldsymbol{\pi}} \left[ \sum_{t=0}^{T-1} \frac{(1 + \sqrt{d_s})\beta_{n-1}(\delta) \left\| \boldsymbol{\sigma}_{n-1}(\boldsymbol{s}_t, \boldsymbol{\pi}(\boldsymbol{s}_t)) \right\|}{\sigma} \right].$$

*Proof.* From Sukhija et al. (2024, Corollary 2.) we have,

$$J_c(\boldsymbol{\pi}, \boldsymbol{f}) - J_c(\boldsymbol{\pi}, \boldsymbol{f}^*) = \mathbb{E}_{\boldsymbol{\tau}^{\boldsymbol{\pi}}} \left[ \sum_{t=0}^{T-1} J_{c,t+1}(\boldsymbol{\pi}, \boldsymbol{f}, \boldsymbol{s}_{t+1}) - J_{c,t+1}(\boldsymbol{\pi}, \boldsymbol{f}, \boldsymbol{s}'_{t+1}) \right],$$

(Expectation w.r.t $\boldsymbol{\pi}$ under true dynamics $\boldsymbol{f}^*$)

$$\text{with } \boldsymbol{s}_{t+1} = \boldsymbol{f}^*(\boldsymbol{s}_t, \boldsymbol{\pi}(\boldsymbol{s}_t)) + \boldsymbol{w}_t,$$
$$\text{and } \boldsymbol{s}'_{t+1} = \boldsymbol{f}(\boldsymbol{s}_t, \boldsymbol{\pi}(\boldsymbol{s}_t)) + \boldsymbol{w}_t.$$

Furthermore, $J_{c,t+1}(\boldsymbol{\pi}, \boldsymbol{f}, \boldsymbol{s}) \in [0, TC_{\max}]$ for all $\boldsymbol{\pi}, \boldsymbol{f}, \boldsymbol{s}$, and $t$. Therefore, given $\boldsymbol{s}_t$,

$$\left| \mathbb{E}_{\boldsymbol{w}_t} \left[ J_{c,t+1}(\boldsymbol{\pi}, \boldsymbol{f}, \boldsymbol{s}'_{t+1}) - J_{c,t+1}(\boldsymbol{\pi}, \boldsymbol{f}, \boldsymbol{s}_{t+1}) \right] \right|$$

$$\leq \max \left\{ \sqrt{\mathbb{E}_{\boldsymbol{w}_t}[J_{c,t+1}^2(\boldsymbol{\pi}, \boldsymbol{f}, \boldsymbol{s}'_{t+1})]}, \sqrt{\mathbb{E}_{\boldsymbol{w}_t}[J_{c,t+1}^2(\boldsymbol{\pi}, \boldsymbol{f}, \boldsymbol{s}_{t+1})]} \right\} \min \left\{ \frac{\|\boldsymbol{f}^*(\boldsymbol{s}_t, \boldsymbol{\pi}(\boldsymbol{s}_t)) - \boldsymbol{f}(\boldsymbol{s}_t, \boldsymbol{\pi}(\boldsymbol{s}_t))\|}{\sigma}, 1 \right\}$$

(Kakade et al., 2020, Lemma C.2.)

$$\leq TC_{\max} \min \left\{ \frac{\|\boldsymbol{f}^*(\boldsymbol{s}_t, \boldsymbol{\pi}(\boldsymbol{s}_t)) - \boldsymbol{f}(\boldsymbol{s}_t, \boldsymbol{\pi}(\boldsymbol{s}_t))\|}{\sigma}, 1 \right\}$$

$$\leq TC_{\max} \min \left\{ \frac{(1 + \sqrt{d_s})\beta_{n-1}(\delta) \|\boldsymbol{\sigma}_{n-1}(\boldsymbol{s}_t, \boldsymbol{\pi}(\boldsymbol{s}_t))\|}{\sigma}, 1 \right\} \quad \text{(Sukhija et al., 2024, Corollary 3)}$$

$$\square$$

From hereon let $C = \frac{(1+\sqrt{d_s}) \max\{R_{\max}, C_{\max}, \sigma_0\}}{\sigma}$.

**Lemma A.7.** *Let assumption $4.1 - 4.5$ hold. Then we have $\forall n, N \geq 0$ with probability at least $1 - \delta$*

$$\max_{\boldsymbol{\pi} \in \mathcal{S}_n} \mathbb{E}_{\boldsymbol{\tau}^{\boldsymbol{\pi}}} \left[ \sum_{t=0}^{T-1} \|\boldsymbol{\sigma}_{N+n-1}(\boldsymbol{s}_t, \boldsymbol{\pi}(\boldsymbol{s}_t))\| \right] \leq T^2 C \frac{\sqrt{d_s}\sigma_0}{\sqrt{\log(1 + \sigma^{-2}\sigma_0^2)}} \sqrt{\frac{\beta_{n+N-1}^2(\delta)\gamma_{n+N-1}(k)}{N}}.$$

*Proof.* Consider any $N > 0$,

$$\max_{\boldsymbol{\pi} \in \mathcal{S}_n} \mathbb{E}_{\boldsymbol{\tau}^{\boldsymbol{\pi}}} \left[ \sum_{t=0}^{T-1} \|\boldsymbol{\sigma}_{N+n-1}(\boldsymbol{s}_t, \boldsymbol{\pi}(\boldsymbol{s}_t))\| \right] \leq \frac{1}{N} \sum_{i=0}^{N-1} \max_{\boldsymbol{\pi} \in \mathcal{S}_n} \mathbb{E}_{\boldsymbol{\tau}^{\boldsymbol{\pi}}} \left[ \sum_{t=0}^{T-1} \|\boldsymbol{\sigma}_{n+i}(\boldsymbol{s}_t, \boldsymbol{\pi}(\boldsymbol{s}_t))\| \right]$$

(Monotonocity of the variance)

$$\leq \frac{1}{N} \sum_{i=0}^{N-1} \max_{\boldsymbol{\pi} \in \mathcal{S}_{n+i}} \mathbb{E}_{\boldsymbol{\tau}^{\boldsymbol{\pi}}} \left[ \sum_{t=0}^{T-1} \|\boldsymbol{\sigma}_{n+i}(\boldsymbol{s}_t, \boldsymbol{\pi}(\boldsymbol{s}_t))\| \right]$$

(Monotonocity of the safe set)

$$= \frac{1}{N} \sum_{i=n}^{n+N-1} \mathbb{E}_{\boldsymbol{\tau}^{\boldsymbol{\pi}_i^*}} \left[ \sum_{t=0}^{T-1} \|\boldsymbol{\sigma}_i(\boldsymbol{s}_t, \boldsymbol{\pi}_i^*(\boldsymbol{s}_t))\| \right]$$

(Definition of $\boldsymbol{\pi}_i^*$)

$$= \frac{1}{N} \sum_{i=n}^{n+N-1} \mathbb{E}_{\boldsymbol{\tau}^{\boldsymbol{\pi}_i}} \left[ \sum_{t=0}^{T-1} \|\boldsymbol{\sigma}_i(\boldsymbol{s}_t, \boldsymbol{\pi}_i(\boldsymbol{s}_t))\| \right] + \frac{1}{N}(J(\boldsymbol{\pi}_i^*) - J(\boldsymbol{\pi}_i))$$

Let $r_i = J_i(\boldsymbol{\pi}_i^*) - J(\boldsymbol{\pi}_i)$, where $\boldsymbol{\pi}_i$ is the policy proposed by Equation (4). We analyze this regret term. Note that since, we optimistically pick dynamics from $\mathcal{M}_n$, we have $J_i(\boldsymbol{\pi}_i^*) \leq J(\boldsymbol{\pi}_i, \boldsymbol{f}_i)$, where $\boldsymbol{f}_i$ are the optimistic dynamics. Therefore, $r_i \leq J(\boldsymbol{\pi}_i, \boldsymbol{f}_i) - J(\boldsymbol{\pi}_i, \boldsymbol{f}^*)$. Hence, we can invoke Lemma A.6 to get

$$r_i \leq TC \left[ \sum_{t=0}^{T-1} \beta_i(\delta) \|\boldsymbol{\sigma}_i(\boldsymbol{s}_t, \boldsymbol{\pi}(\boldsymbol{s}_t))\| \right].$$

Therefore,

$$
\begin{aligned}
\max_{\boldsymbol{\pi} \in \mathcal{S}_n} \mathbb{E}_{\boldsymbol{\tau}^{\boldsymbol{\pi}}} \left[ \sum_{t=0}^{T-1} \| \boldsymbol{\sigma}_{N+n-1}(\boldsymbol{s}_t, \boldsymbol{\pi}(\boldsymbol{s}_t)) \| \right] &\leq \frac{TC\beta_{n+N-1}(\delta)}{N} \sum_{i=n}^{n+N-1} \mathbb{E}_{\boldsymbol{\tau}^{\boldsymbol{\pi}_i}} \left[ \sum_{t=0}^{T-1} \| \boldsymbol{\sigma}_i(\boldsymbol{s}_t, \boldsymbol{\pi}_i(\boldsymbol{s}_t)) \| \right] \\
&\leq \frac{TC\beta_{n+N-1}(\delta)}{N} \sum_{i=n}^{n+N-1} \mathbb{E}_{\boldsymbol{\tau}^{\boldsymbol{\pi}_i}} \left[ \sum_{t=0}^{T-1} \| \boldsymbol{\sigma}_i(\boldsymbol{s}_t, \boldsymbol{\pi}_i(\boldsymbol{s}_t)) \| \right] \\
&\leq \frac{TC\beta_{n+N-1}(\delta)}{N} \sqrt{NT} \sqrt{\sum_{i=n}^{n+N-1} \mathbb{E}_{\boldsymbol{\tau}^{\boldsymbol{\pi}_i}} \left[ \sum_{t=0}^{T-1} \| \boldsymbol{\sigma}_i(\boldsymbol{s}_t, \boldsymbol{\pi}_i(\boldsymbol{s}_t)) \|^2 \right]} \\
&\qquad\qquad\qquad\qquad \text{(Cauchy-Schwartz)} \\
&\leq \frac{TC\beta_{n+N-1}(\delta)}{N} \sqrt{NT} \sqrt{\sum_{i=0}^{n+N-1} \mathbb{E}_{\boldsymbol{\tau}^{\boldsymbol{\pi}_i}} \left[ \sum_{t=0}^{T-1} \| \boldsymbol{\sigma}_i(\boldsymbol{s}_t, \boldsymbol{\pi}_i(\boldsymbol{s}_t)) \|^2 \right]} \\
&\leq \frac{TC\frac{\sqrt{Td_s}\sigma_0}{\sqrt{\log(1+\sigma^{-2}\sigma_0^2)}}\beta_{n+N-1}(\delta)}{N} \sqrt{NT} \sqrt{\gamma_{n+N-1}(k)} \\
&\qquad\qquad\qquad\qquad \text{(Curi et al., 2020, Lemma 17)} \\
&= T^2 C \frac{\sqrt{d_s}\sigma_0}{\sqrt{\log(1+\sigma^{-2}\sigma_0^2)}} \sqrt{\frac{\beta_{n+N-1}^2(\delta)\gamma_{n+N-1}(k)}{N}}
\end{aligned}
$$

$\square$

**Lemma A.8.** *Let assumption 4.1 – 4.5 hold and define $N_n$ to be the smallest integer such that*

$$
T^3 C^2 \frac{\sqrt{d_s}\sigma_0}{\sqrt{\log(1+\sigma^{-2}\sigma_0^2)}} \beta_{n+N_n-1}^2(\delta) \sqrt{\frac{\gamma_{n+N_n-1}(k)}{N_n}} \leq \epsilon.
$$

*Then, we have $\forall \boldsymbol{\pi} \in \mathcal{S}_n$, $\boldsymbol{f} \in \mathcal{M}_{n+N_n-1}$ with probability at least $1-\delta$*

$$
|J_c(\boldsymbol{\pi}, \boldsymbol{f}) - J_c(\boldsymbol{\pi}, \boldsymbol{f}^*)| \leq \epsilon, \text{ and, } |J_r(\boldsymbol{\pi}, \boldsymbol{f}) - J_r(\boldsymbol{\pi}, \boldsymbol{f}^*)| \leq \epsilon.
$$

*Proof.*

$$
\begin{aligned}
|J_c(\boldsymbol{\pi}, \boldsymbol{f}) - J_c(\boldsymbol{\pi}, \boldsymbol{f}^*)| &\leq TC_{\max} \mathbb{E}_{\boldsymbol{\tau}^{\boldsymbol{\pi}}} \left[ \sum_{t=0}^{T-1} \frac{(1+\sqrt{d_s})\beta_{n+N_n-1}(\delta) \| \boldsymbol{\sigma}_{n+N_n-1}(\boldsymbol{s}_t, \boldsymbol{\pi}(\boldsymbol{s}_t)) \|}{\sigma} \right] \\
&\qquad\qquad\qquad\qquad \text{(Lemma A.6)} \\
&\leq TC\beta_{n+N_n-1} \left[ \sum_{t=0}^{T-1} \| \boldsymbol{\sigma}_{n+N_n-1}(\boldsymbol{s}_t, \boldsymbol{\pi}(\boldsymbol{s}_t)) \| \right] \\
&\leq TC\beta_{n+N_n-1} T^2 C \frac{\sqrt{d_s}\sigma_0}{\sqrt{\log(1+\sigma^{-2}\sigma_0^2)}} \beta_{n+N_n-1}(\delta) \sqrt{\frac{\gamma_{n+N_n-1}(k)}{N_n}} \\
&\qquad\qquad\qquad\qquad \text{(Lemma A.7)} \\
&\leq \epsilon
\end{aligned}
$$

We can apply the same inequalities for $J_r$. $\square$

**Corollary A.9.** *Let assumption 4.1 – 4.5 hold. Consider any $n \geq 0$ and define $N_n$ as in Lemma A.8. Then we have with probability at least $1-\delta$*

$$
\mathcal{S}_{n+N_n} \supseteq \mathcal{R}^\varepsilon(\mathcal{S}_n).
$$

*Proof.* From Lemma A.8, we have $\forall \boldsymbol{\pi} \in \mathcal{S}_n$, $\boldsymbol{f} \in \mathcal{M}_{n+N_n-1}$, $|J_c(\boldsymbol{\pi}, \boldsymbol{f}) - J_c(\boldsymbol{\pi}, \boldsymbol{f}^*)| \leq \epsilon$, therefore $P_{n+N_n-1}(\boldsymbol{\pi}) \leq J_c(\boldsymbol{\pi}, \boldsymbol{f}^*) + \epsilon$. For the sake of contradiction, assume there exists a policy $\boldsymbol{\pi} \in \mathcal{R}^\varepsilon(\mathcal{S}_n) \setminus \mathcal{S}_{n+N_n}$. We study the case where $\boldsymbol{\pi} \in \mathcal{R}^\varepsilon(\mathcal{S}_n) \setminus \mathcal{S}_n$ else we have a contradiction $(\mathcal{S}_n \subseteq \mathcal{S}_{n+N_n})$. Since $\boldsymbol{\pi} \in \mathcal{R}^\varepsilon(\mathcal{S}_n) \setminus \mathcal{S}_n$, there exists a $\boldsymbol{\pi}' \in \mathcal{S}_n$ such that $J_c(\boldsymbol{\pi}') + D(\boldsymbol{\pi}, \boldsymbol{\pi}') \leq d - \epsilon$ (see Theorem 4.8). Hence, we get

$$d \geq J_c(\boldsymbol{\pi}') + \epsilon + D(\boldsymbol{\pi}, \boldsymbol{\pi}')$$
$$\geq P_{n+N_n-1}(\boldsymbol{\pi}') + D(\boldsymbol{\pi}, \boldsymbol{\pi}').$$

Since, $\boldsymbol{\pi}' \in \mathcal{S}_n \subseteq \mathcal{S}_{n+N_n-1}$, by the definition of the safe set (c.f. Definition 4.7), this implies that $\boldsymbol{\pi} \in \mathcal{S}_{n+N_n}$, which is a contradiction. $\square$

A key property of $N_n$ is that it increases monotonously with $n$. Moreover, for a given $n \geq 0$, $N_n$ is the smallest integer satisfying

$$N_n \geq \frac{\gamma_{n+N_n-1}(k)\beta_{n+N_n-1}^4(\delta)T^6 C^4 \frac{d_s\sigma_0^2}{\log(1+\sigma^{-2}\sigma_0^2)}}{\epsilon^2}.$$

Both functions $n \mapsto \gamma_n$, and $n \mapsto \beta_n$ are monotonically increasing with $n$. Hence increasing $n$, increases the right-hand side of the inequality, and therefore $N_n$.

**Lemma A.10.** *Let assumption 4.1 – 4.5 hold and consider $n^* \geq (H+1)N_{n^*}$. Then we have with probability at least $1 - \delta$ for all $n \geq n^*$*

$$\mathcal{S}_n \supseteq \mathcal{R}_H^\varepsilon(\mathcal{S}_0).$$

*Proof.* To prove this, we show for any positive integer $k \leq H$, that $\mathcal{S}_{kN_{n^*}} \supseteq \mathcal{R}_k^\varepsilon(\mathcal{S}_0)$ by induction. Moreover for any $k$, let $T_k = T_{k-1} + N_{T_{k-1}}$ and $T_0 = 0$. We inductively show that $T_k \leq kN_{n^*}$ for all $k \leq H$.

For the base case $k = 1$, we have $T_1 = N_0 \leq N_{n^*}$ since $n^* \geq 0$. Consider any $k \leq H$, then, we have $T_k = T_{k-1} + N_{T_{k-1}}$. By induction $T_{k-1} \leq (k-1)N_{n^*}$. Therefore, $T_k \leq (k-1)N_{n^*} + N_{(k-1)N_{n^*}}$. Furthermore, note that $(k-1)N_{n^*} \leq n^*$ for all $k \leq H$. Therefore, $T_k \leq (k-1)N_{n^*} + N_{n^*} = kN_{n^*}$. Next, we have from Corollary A.9, $\mathcal{S}_{T_k} \supseteq \mathcal{R}^\varepsilon(\mathcal{S}_{T_{k-1}}) := \mathcal{R}_k^\varepsilon(\mathcal{S}_0)$. Moreover, $\mathcal{S}_{T_1} := \mathcal{S}_{N_0} \supseteq \mathcal{R}^\varepsilon(\mathcal{S}_0)$. Similarly, $\mathcal{S}_{T_2} := \mathcal{S}_{N_0+N_{N_0}} \supseteq \mathcal{R}^\varepsilon(\mathcal{S}_1) := \mathcal{R}_2^\varepsilon(\mathcal{S}_0)$, etc. Therefore, we get $\mathcal{S}_{HN_{n^*}} \supseteq \mathcal{S}_{T_H} \supseteq \mathcal{R}_H^\varepsilon(\mathcal{S}_0)$. As $n^* \geq HN_{n^*}$, this completes the proof. $\square$

**Lemma A.11.** *Let assumption 4.1 – 4.5 hold and consider the smallest integer $n^*$ such that*

$$\frac{n^*}{\gamma_{n^*}(k)\beta_{n^*}^4(\delta)} \geq \frac{(H+1)T^6 C^4 \frac{d_s\sigma_0^2}{\log(1+\sigma^{-2}\sigma_0^2)}}{\epsilon^2}. \tag{10}$$

*Then we have for all $n \geq n^*$*

$$\mathcal{S}_n \supseteq \mathcal{R}_H^\varepsilon(\mathcal{S}_0).$$

*Moreover, we have for all $n \geq n^*$, $\boldsymbol{\pi} \in \mathcal{R}_H^\varepsilon(\mathcal{S}_0)$ that $|J_c(\boldsymbol{\pi}, \boldsymbol{f}) - J_c(\boldsymbol{\pi}, \boldsymbol{f}^*)| \leq \epsilon$ and $|J_r(\boldsymbol{\pi}, \boldsymbol{f}) - J_r(\boldsymbol{\pi}, \boldsymbol{f}^*)| \leq \epsilon$.*

*Proof.* Note that for any $n$, $N_n$ is defined as the smallest integer satisfying:

$$\frac{N_n}{\gamma_{n+N_n-1}(k)\beta_{n+N_n-1}^4(\delta)} \geq \frac{T^6 C^4 \frac{d_s\sigma_0^2}{\log(1+\sigma^{-2}\sigma_0^2)}}{\epsilon^2}.$$

From Lemma A.10, for $n^* = (H+1)N_{n^*}$, we have for all $n \geq n^*$

$$\mathcal{S}_n \supseteq \mathcal{R}_H^\varepsilon(\mathcal{S}_0).$$

We show that the solution to Equation (5) satisfies this condition. Moreover, let $n^* = (H+1)N_{n^*}$

$$\frac{N_{n^*}}{\gamma_{n^*+N_{n^*}-1}(k)\beta_{n^*+N_{n^*}-1}^4(\delta)} = \frac{\frac{n^*}{H+1}}{\gamma_{n^*+\frac{n^*}{H+1}-1}(k)\beta_{n^*+\frac{n^*}{H+1}-1}^4(\delta)}$$

$$\geq \frac{\frac{n^*}{H+1}}{\gamma_{n^*}(k)\beta_{n^*}^4(\delta)}$$

Picking $n^*$ as the smallest integer satisfying

$$\frac{n^*}{\gamma_{n^*}(k)\beta_{n^*}^4(\delta)} \geq \frac{(H+1)T^6C^4\frac{d_s\sigma_0^2}{\log(1+\sigma^{-2}\sigma_0^2)}}{\epsilon^2},$$

ensures that

$$\frac{N_{n^*}}{\gamma_{n^*+N_{n^*}-1}(k)\beta_{n^*+N_{n^*}-1}^4(\delta)} \geq \frac{T^6C^4\frac{d_s\sigma_0^2}{\log(1+\sigma^{-2}\sigma_0^2)}}{\epsilon^2}$$

Finally, from Lemma A.8 we have that $S_{HN_{n^*}} \supseteq \mathcal{R}_H^\varepsilon(\mathcal{S}_0)$.

Therefore, for all $n \geq n^*$, $\boldsymbol{\pi} \in \mathcal{R}_H^\varepsilon(\mathcal{S}_0)$, $\boldsymbol{f} \in \mathcal{M}_{n^*+N_{n^*}-1}$ with probability at least $1 - \delta$

$$|J_c(\boldsymbol{\pi}, \boldsymbol{f}) - J_c(\boldsymbol{\pi}, \boldsymbol{f}^*)| \leq \epsilon, \text{ and, } |J_r(\boldsymbol{\pi}, \boldsymbol{f}) - J_r(\boldsymbol{\pi}, \boldsymbol{f}^*)| \leq \epsilon.$$

$\square$

*Proof of Theorem 4.8.* We prove in Corollary A.5, that all policies in $\mathcal{S}_n$ are safe for all $n \geq 0$. ACTSAFE is safe since it picks policies only from $\mathcal{S}_n$.

For optimality, we showed in Lemma A.10 for all $n \geq n^*$ that $S_n \supseteq \mathcal{R}_H^\varepsilon(\mathcal{S}_0)$. Moreover, we have $\forall \boldsymbol{\pi} \in \mathcal{R}_H^\varepsilon(\mathcal{S}_0)$, $\boldsymbol{f} \in \mathcal{M}_{n^*+N_{n^*}-1} \supseteq \mathcal{M}_n, |J_r(\boldsymbol{\pi}, \boldsymbol{f}) - J_r(\boldsymbol{\pi}, \boldsymbol{f}^*)| \leq \epsilon$. Let $\boldsymbol{\pi}^*$ be the optimal policy and let $\tilde{\boldsymbol{\pi}}_n$ denote the solution to $\arg\max_{\boldsymbol{\pi} \in \mathcal{S}_n} \min_{\boldsymbol{f} \in \mathcal{M}_n} J_r(\boldsymbol{\pi}, \boldsymbol{f})$. For the sake of contradiction, assume that

$$J_r(\tilde{\boldsymbol{\pi}}_n) < \max_{\boldsymbol{\pi} \in \mathcal{R}_H^\epsilon(\mathcal{S}_0)} J_r(\boldsymbol{\pi}, \boldsymbol{f}^*) - \epsilon. \tag{11}$$

Furthermore, let $P_n^r(\boldsymbol{\pi}) = \min_{\boldsymbol{f} \in \mathcal{M}_n} J_r(\boldsymbol{\pi}, \boldsymbol{f})$ for all $\boldsymbol{\pi} \in \Pi$.

$$\begin{aligned}
P_n^r(\boldsymbol{\pi}^*) &\leq \max_{\boldsymbol{\pi} \in \mathcal{S}_n} P_n^r(\boldsymbol{\pi}) \\
&= P_n^r(\tilde{\boldsymbol{\pi}}_n) \\
&\leq J_r(\tilde{\boldsymbol{\pi}}_n) \\
&< J_r(\boldsymbol{\pi}^*, \boldsymbol{f}^*) - \epsilon && \text{(contradiction assumption)} \\
&\leq P_n^r(\boldsymbol{\pi}^*). && \text{(Lemma A.11)}
\end{aligned}$$

This is a contradiction, which completes the proof.

$\square$

# B EXPERIMENT DETAILS

## B.1 GP EXPERIMENTS

For the GP experiments, we approximate Equation (7) with the following unconstrained optimization problem.

$$\arg\max_{\boldsymbol{\pi}\in\Pi}\max_{\boldsymbol{f}\in\mathcal{Q}_n} J_n(\boldsymbol{\pi},\boldsymbol{f}) - \lambda\max\left\{\max_{\boldsymbol{f}'\in\mathcal{Q}_n} J_c(\boldsymbol{\pi},\boldsymbol{f}') - d, 0\right\}. \tag{12}$$

Here $\lambda$ is a (large) penalty term that is used to discourage constraint violation. We use the iCEM (Pinneri et al., 2021) optimizer to solve the constrained optimization above. Effectively, given a sequence of actions $\{\boldsymbol{a}_t\}_{t=0}^H$, we roll them out on our learned GP model using the TS1 approach from Chua et al. (2018). Moreover, we maintain $P$ particles, and given the state $(\boldsymbol{s}_t^p, \boldsymbol{a}_t^p)$ for the $p$-th particle, we determine the next state $\boldsymbol{s}_{t+1}^p$, by sampling from $\mathcal{N}(\boldsymbol{\mu}_n(\boldsymbol{s}_t^p, \boldsymbol{a}_t^p), \boldsymbol{\sigma}_n(\boldsymbol{s}_t^p, \boldsymbol{a}_t^p))$. Accordingly, for each action sequence $\{\boldsymbol{a}_t\}_{t=0}^H$, we obtain $P$ trajectories and we empirically solve $\max_{\boldsymbol{f}'\in\mathcal{Q}_n} J_c(\boldsymbol{\pi},\boldsymbol{f}')$ by taking the max over the $P$ trajectories. This approach is also proposed by Kakade et al. (2020) as a heuristic for optimizing over the dynamics.

**Rewards and constraints** The reward function is designed to penalize deviations in both the angular position and the control input from the target behavior. For both the PENDULUM and CARTPOLE, the state of the pole can be defined as follows. Let $\theta$ be the current angle, $\omega$ the angular velocity, and $u$ the control input. The target angle is denoted as $\theta_{\text{target}}$, and the angular error between the current angle and the target angle is $\Delta\theta$. The reward and cost functions for the PENDULUM environment are given by

$$r_{\text{Pendulum}} = -\left(\Delta\theta^2 + 0.1\cdot\omega^2 + 0.02\cdot u^2\right), \quad c_{\text{Pendulum}} = \max\{|\omega| - 6.0, 0.0\}, d = 0.0.$$

For the CARTPOLE environment, the position and velocity of the slider are defined as $p$ and $v$ respectively. The reward for the CARTPOLE environment is the given by

$$r_{\text{Cartpole}} = -\left(\Delta\theta^2 + p^2 + 0.1\cdot\left(v^2 + \omega^2\right)\right) - 0.01\cdot u^2, \quad c_{\text{Cartpole}} = \max\{|p| - 0.5, 0.0\}, d = 0.75.$$

## B.2 VISION CONTROL EXPERIMENTS

We provide an open-source implementation of our experiments in `https://github.com/yardenas/actsafe`. We encourage readers to use it, as it contains additional important implementation details. In all the experiments below, our policy consists of 750K parameters, a several orders of magnitudes compared to previous works on provable safe explorations.

**Approximating Equation (7)** We solve the constraint optimization problem in Equation (7) using the LBSGD solver from Usmanova et al. (2024). LBSGD is a first-order optimizer that uses a logarithmic barrier function to enforce constraint satisfaction. Previous works from Ni & Kamgarpour (2024); As et al. (2024) have successfully applied LBSGD for planning in model-based RL with CMDPs, showing notably fewer constraint violations than alternative solvers like the augmented Lagrangian method (As et al., 2022).

To approximate Equation (7), we maintain an RSSM ensemble of $P$ particles and given the state action pair $(\boldsymbol{s}_t, \pi_n(\boldsymbol{a}_t|\boldsymbol{s}_t))$, we obtain $P$ estimates $\{\boldsymbol{s}_{t+1}^p\}_{p=1}^P$ for the next state. We estimate $\boldsymbol{\sigma}_n^2$ with the variance/disagreement between the ensemble members, i.e., $\text{Var}\left(\{\boldsymbol{s}_{t+1}^p\}_{p=1}^P\right)$. We obtain the next state $\boldsymbol{s}_{t+1}$ by uniform sampling from $\{\boldsymbol{s}_{t+1}^p\}_{p=1}^P$, i.e., TS1 from Chua et al. (2018). Akin to Yu et al. (2020), we approximate $\max_{\boldsymbol{f}'\in\mathcal{Q}_n} J_c(\boldsymbol{\pi},\boldsymbol{f}')$ by penalizing the cost function with $\boldsymbol{\sigma}_n$

$$J_{c-\lambda\boldsymbol{\sigma}}(\boldsymbol{\pi}_n) = \mathbb{E}_{\boldsymbol{\pi}_n}\left[\sum_{t=0}^H \gamma^t(c(\boldsymbol{s}_t, \boldsymbol{a}_t) + \lambda\left\|\boldsymbol{\sigma}_n(\boldsymbol{s}_t, \boldsymbol{a}_t)\right\|)\right],$$

where $\lambda$ is a pessimism parameter. Yu et al. (2020) show that for an appropriate choice of $\lambda$, $J_{c-\lambda\boldsymbol{\sigma}}(\boldsymbol{\pi}_n)$ is indeed a pessimistic estimate of $J_c(\boldsymbol{\pi}_n)$. However, in our experiments we treat $\lambda$ as a hyper-parameter.

**Safety experiments** We focus on SAFETY-GYM to showcase our practical algorithm design maintains constraint satisfaction during learning. Our experiments rely on a newer fork of SAFETY-GYM which is available via our open-source code. We follow the experimental setup of Ray et al. (2019); As et al. (2022) and an episode length of $T = 1000$. We set the cost budget for each episode

to $d = 25$ for SAFETY-GYM (see Ray et al., 2019). After each training epoch we estimate $J_r(\boldsymbol{\pi}_n)$ and $J_c(\boldsymbol{\pi}_n)$ by sampling 50 episodes, denoting the estimates with $\hat{J}_r$ and $\hat{J}_c$. Unless specified otherwise, in all our experiments we use 5 random seeds and report the median and standard error across these seeds. Finally, we use a budget of 5M training steps for each training run. To make a fair comparison with As et al. (2022); Huang et al. (2024), we fix the ratio of environment steps and update steps of the model and policy. While Huang et al. (2024) use the RSSM model from Hafner et al. (2023), our implementation uses the (older) one from Hafner et al. (2019) and As et al. (2022).

**Sparse SAFETY-GYM** Let $d_t^{\mathrm{RG}}$ be the euclidean distance between the robot and the goal/button at time step $t$, $d_t^{\mathrm{BG}}$ the distance between the box and the goal position and $d_t^{\mathrm{RB}}$ the distance between the robot and the box positions. Furthermore, denote $\mathrm{tol}(x, l, u)$ as the tolerance function from Tassa et al. (2018), where $l, u$ denotes lower and upper bounds respectively.

| Environment | Dense Reward | Sparse Reward |
|---|---|---|
| GOTOGOAL | $d_{t-1}^{\mathrm{RG}} - d_t^{\mathrm{RG}} + \mathbf{1}_{d_t^{\mathrm{BG}} \leq 0.3}$ | $\mathrm{tol}(d_t^{\mathrm{RG}}, 0, 0.45) \cdot (d_{t-1}^{\mathrm{RG}} - d_t^{\mathrm{RG}}) + \mathbf{1}_{d_t^{\mathrm{RG}} \leq 0.3}$ |
| PRESSBUTTON | $d_{t-1}^{\mathrm{RG}} - d_t^{\mathrm{RG}} + \mathbf{1}_{d_t^{\mathrm{BG}} \leq 0.3}$ | $\mathbf{1}_{d_t^{\mathrm{RG}} \leq 0.1}$ |
| PUSHBOX | $d_{t-1}^{\mathrm{RB}} - d_t^{\mathrm{RB}} + d_{t-1}^{\mathrm{BG}} - d_t^{\mathrm{BG}} + \mathbf{1}_{d_t^{\mathrm{BG}} \leq 0.3}$ | $\mathrm{tol}(d_t^{\mathrm{RB}}, 0, 0.5) \cdot (d_{t-1}^{\mathrm{RB}} - d_t^{\mathrm{RB}}) + d_{t-1}^{\mathrm{BG}} - d_t^{\mathrm{BG}} + \mathbf{1}_{d_t^{\mathrm{BG}} \leq 0.3}$ |

Table 1: Comparison of the reward functions in the base environments of SAFETY-GYM and our sparse rewards environments.

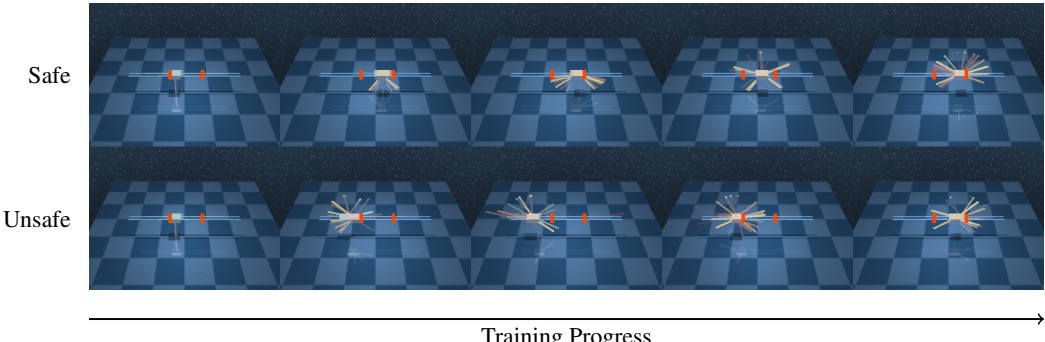

Training Progress

Figure 7: CARTPOLE environment as an example of a problem instance of safe exploration. Each scene summarizes a trajectory that was collected in increasing training iterations. The agent incurs a cost whenever the cart goes outside of the area between the two red vertical lines. The goal is to learn a policy that swings the pole to the top position, while ensuring the expected accumulated cost is bounded *during learning*. Learning in this setting is much more challenging, as agents can only try out control policies that known to be safe.

**Cartpole exploration** In this task, the agent receives a sparse reward when it swings up a pendulum to the top position and when the slider (a.k.a cart) is centered. The RWRL benchmark (Dulac-Arnold et al., 2019) adds a safety constraint that enforces the slider to remain in a certain distance from the center (see Figure 7). As in Dulac-Arnold et al. (2019), we use a cost budget of $d = 100$ and an episode length of $T = 1000$ steps. Adding the safety constraint adds a significant challenge, as any safe policy is much more limited in exploration. In addition to the safety constraint, we add a cost for taking actions, as done in Curi et al. (2020). Combining all these factors together, makes a challenging exploration task, as we show in our experiments. Further implementation details can be found in our open-source code.

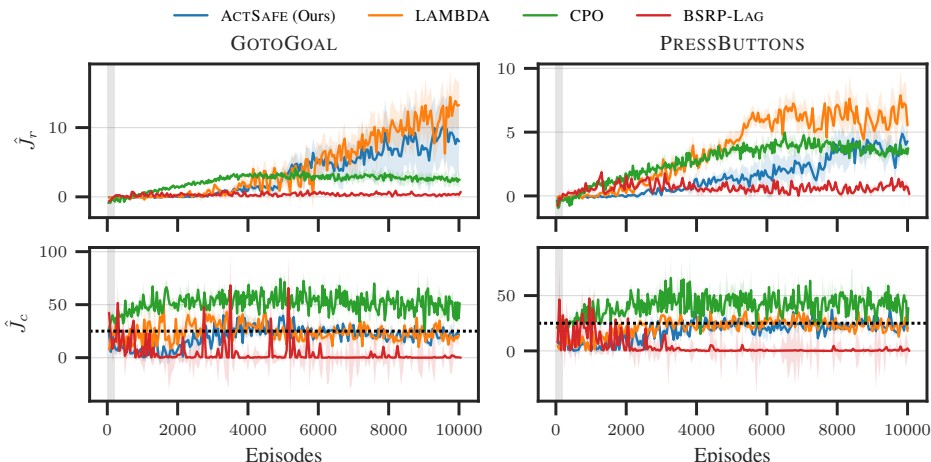

Figure 8: Performance and safety in with the DOGGO robot.

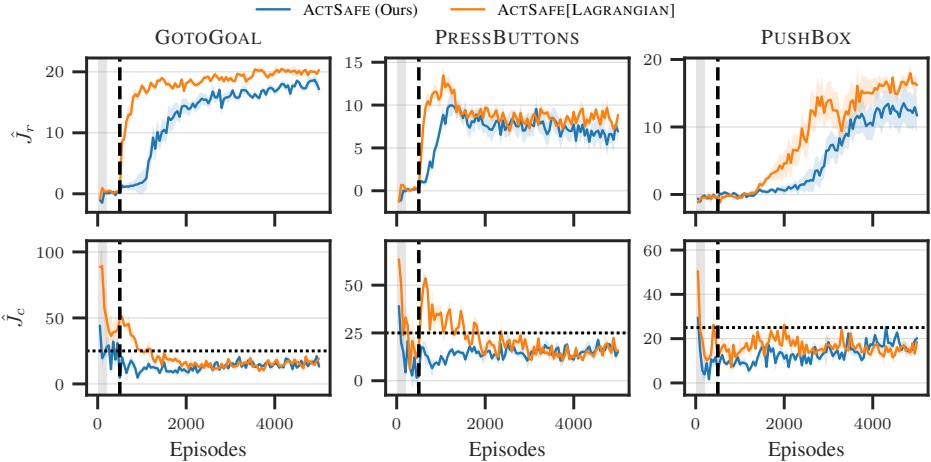

Figure 9: Augmented Lagrangian from As et al. (2022) compared to LBSGD of Usmanova et al. (2024). LBSGD significantly reduces the number of unsafe episodes.

## C   ADDITIONAL EXPERIMENTS

**Experiments with the DOGGO Robot**   In this experiment we compare ACTSAFE with the same baseline algorithms from Section 5 on SAFETY-GYM's DOGGO robot. We omit our results in the PUSHBOX environment as all baselines failed to solve it. As shown in Figure 8, similarly to the results in Section 5, ACTSAFE maintains safety during learning, while moderately underperforming LAMBDA. Overall, ACTSAFE outperforms CPO both in terms of safety and performance and BSRP-LAG of Huang et al. (2024) in terms of performance.

**Ablating LBSGD**   One assumptions of LBSGD that we cannot formally satisfy relates to unbiasedness of the evaluation of the objective, constraints and their gradients. In principle, satisfying this assumption will allow us to guarantee that all iterates of Equation (7) are feasible, i.e., satisfy the pessimistic constraint. This is in contrast to primal-dual methods, such as the Augmented Lagrangian of As et al. (2022) that lacks any guarantees on feasibility during optimization. While it is hard to formally satisfy LBSGD's unbiasedness assumption, we empirically observe that LBSGD allows us to keep constraint satisfaction during learning. We present this result in Figure 9. As shown, even after initializing both variants with initial data from the burn-in period, ACTSAFE[LAGRANGIAN] fails to satisfy the constraints throughout learning. As in the main results on safety in Figure 4, compared to Augmented Lagrangian, LBSGD maintains safety during learning at a slight price of performance.

**Safe Adaptation**    Here, instead of the warm-up period of data collection, we study the effect of first training on a "safe" environment, like a simulator, and then continuing training on a similar environment, but with shifted dynamics. To this end, we extend GOTOGOAL from SAFETY-GYM to two additional tasks, in which we change the motor gear and floor damping coefficients. The agent is first allowed to explore the "sim" environment for 300K interaction steps before being deployed on the "real" environment. We analyze the impact of our LBSGD optimizer and of pessimism in handling constraint violation during deployment. As shown in Figure 10, without LBSGD and pessimism, ACTSAFE does not always transfer safely to the deployment environment. Furthermore, intuitively, while pessimism is crucial for maintaining safety while adapting to distribution shifts, it may sometimes hinder performance of the main objective. This experiment demonstrates that, if one has no initial data, one can use ACTSAFE in combination with a simulator to achieve safe exploration in practice, with a clear tradeoff of the simulator's fidelity and the degree of pessimism in ACTSAFE.

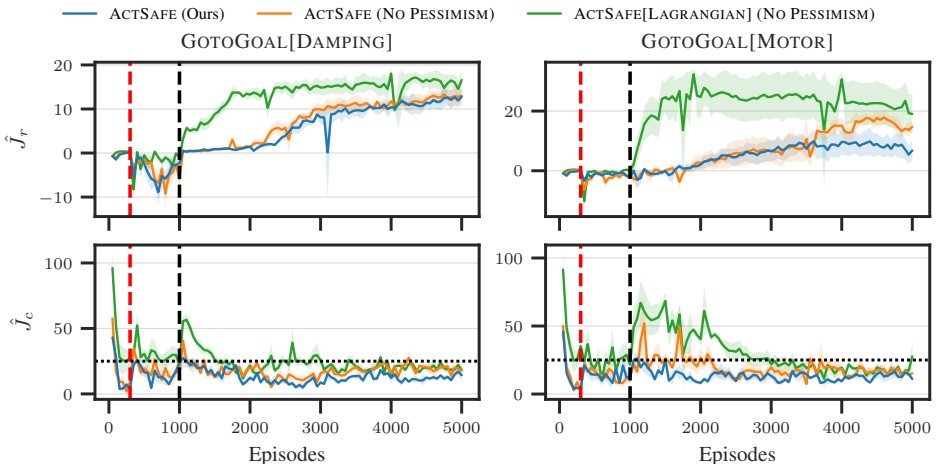

Figure 10: Adaptation to domain shifts. The red dashed vertical line represents the step after which we switch dynamics. Black dashed vertical line represents changing from active exploration to greedily maximizing the reward. We report the mean metrics across 5 seeds.

**Humanoid Proof-of-Concept**    We further demonstrate the scalability of ACTSAFE on the HU-MANOIDBENCH benchmark (Sferrazza et al., 2024). We use a robust, low-level walking policy provided with the benchmark, and input visual observations from a third-person camera view. We compare ACTSAFE with OPAX (Sukhija et al., 2024) on the POLE task, where a humanoid robot must navigate through a field of pole obstacles, as illustrated in Figure 11. In this task, the agent incurs a cost of 1 for each pole it hits and when it falls, while the reward is based on the robot's forward

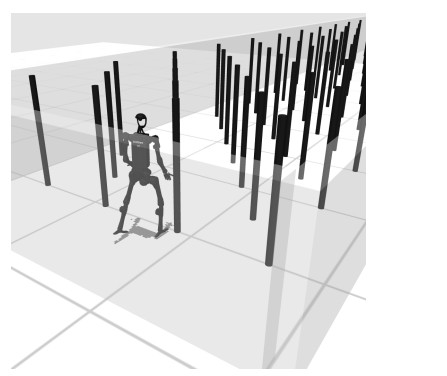

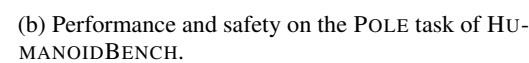

(a) POLE task of HUMANOIDBENCH. The robot has to cross to the other side of the maze while avoiding hitting the poles.

(b) Performance and safety on the POLE task of HU-MANOIDBENCH.

Figure 11: Overview of the Pole task and its performance metrics.

velocity. As shown in Figure 11, ACTSAFE significantly reduces the number of constraint violations compared to OPAX, while maintaining competitive performance on the objective.

**Comparison with OPAX on CARTPOLE** We compare ACTSAFE with OPAX (Sukhija et al., 2023) on the CARTPOLESWINGUPSPARSE task from Section 5.2. Both ACTSAFE and OPAX rely on intrinsic rewards for exploration and model learning, however, ACTSAFE only considers policies from within the pessimistic safe set. We compare ACTSAFE with OPAX trained for 1M and 1.25M steps of pure exploration. ACTSAFE uses 1M exploration steps, as in Section 5.2. As shown in Figure 12, OPAX fails to sufficiently explore the dynamics within 1M steps. The reason being that ACTSAFE can explore in a much more confined state-action space, and therefore visits states with non-zero rewards quicker. This is in contrast to OPAX

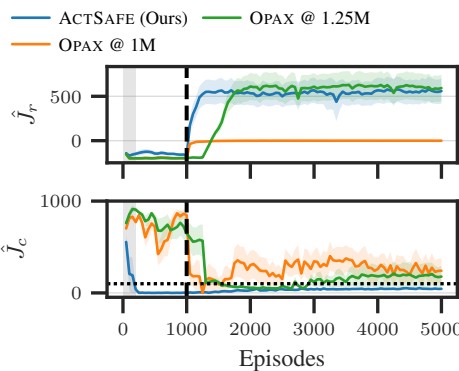

Figure 12: Comparison of ACTSAFE and OPAX in the CARTPOLESWINGUPSPARSE task of RWRL.

which is permitted to explore unsafe action-states as well, and therefore less likely to visit these states within the given training budget. We note that a result in a similar spirit has been observed by Widmer et al. (2023, Figure 2). While 1M steps are not enough for OPAX to fully learn the dynamics when no constraints are imposed on the policy, in Figure 12 we show that after having explored the dynamics for 1.25M steps, OPAX is able to recover an optimal policy. Unsurprisingly, in both experiments OPAX fails to satisfy the constraints, as it optimizes only for the intrinsic reward.

