# OpenReview forum: "ActSafe: Active Exploration with Safety Constraints for Reinforcement Learning"
_ICLR.cc/2025/Conference — ICLR 2025 Poster_

### Official Review · Reviewer_XLTg · 2024-10-31

**Soundness:** 3
**Presentation:** 4
**Contribution:** 3
**Rating:** 6
**Confidence:** 3

**Summary:**

This paper introduces ActSafe, a model-based algorithm for addressing the challenge of safe and efficient exploration in continuous state-action spaces, and provides a detailed theoretical analysis of it. This algorithm integrates OPAX and LBSGD with typical model-based RL methods to achieve safe and efficient exploration performance in vision control tasks. They conduct sufficient experiments in Safety-Gym domain, demonstrating that their approach outperforms prior works regarding safe exploration.

**Strengths:**

1. The problem is well-motivated.
2. This paper is mostly well-written and has solid theoretical foundations. The code provided is very helpful for the reproducibility and review of this paper.
3. The empirical results in sparse reward tasks are impressive.

**Weaknesses:**

1. The proposed algorithm ActSafe, is a combination of  OPAX, LBSGD, and Dreamer(or other model-based RL methods). The technical contributions are incremental.   But considering the experiments are solid and sound, that should be fine.
2. Inconsistency. This algorithm addresses the problem of safe exploration by maintaining a 'safe set' of policies. However, after checking the code provided, there is no such concept of a 'safe set', they just optimize the actor with the typical safe actor-critic method. Since the theoretical results might be unconvincing because of the inconsistency between the theory and practical implementation, it would be nice if the authors could explain this.
3. Baselines.  In Figure 5, the author compares ActSafe with existing methods like OPTIMISTIC、UNIFORM and GREEDY. Are there any other more competitive baselines?  Comparison with methods like GoSafeOpt、OPAX might be necessary.

**Questions:**

1. It would be great if the author could address the main weaknesses I have outlined above. If they are properly addressed, I would be happy to raise my score, as I may have misunderstood the paper.
2. Considering that there are no hard constraints on safety during learning in practical implementations, can ActSafe maintain safety during training after removing the initial offline data?
3. I am very familiar with BSRP-Lag, but the algorithm's performance in the PUSHBOX task doesn't quite match my experience, can you disclose more experimental details?

---

> ### Author Response · Authors · 2024-11-15
> **Response to Reviewer XLTg**
>
> Thank you for your feedback and for acknowledging our theoretical and empirical contributions! Below we address the points raised by the reviewer.
>
> > The proposed algorithm ActSafe, is a combination of OPAX, LBSGD, and Dreamer(or other model-based RL methods).
>
> As mentioned, we indeed use these methods as building blocks of the practical implementation of ActSafe. In particular, we use Dreamer for modeling the latent dynamics and LBSGD as the policy optimizer. These state-of-the-art methods enable efficient scaling of our algorithm to complex visual control tasks. However, our method itself is agnostic to the choice of modeling, i.e., Dreamer or the CMDP solver (LBSGD). We illustrate this in our GP experiments where we use Gaussian processes to model the dynamics and an MPC solver. Furthermore, we leverage insights from OPAX for algorithmic development. However, we would like to highlight that while OPAX gives convergence results, it does not consider safety at all.
> Providing convergence results in our setting is much more challenging as the space of policies over which we search is growing with each iteration. To this end, guarantees for Safe RL methods have been mostly limited to works such as SafeOpt, GoSafeOpt, etc. which do not scale to the high-dimensional setting considered in this work. In this work, we bridge this gap by providing solid theoretical results, as also highlighted by the reviewer, for the more scalable model-based RL setting. Therefore, while we leverage the insights from OPAX, our technical analysis differs significantly from the OPAX algorithm due to the added complexity of safe learning.
>
> > This algorithm addresses the problem of safe exploration by maintaining a 'safe set' of policies. However, after checking the code provided, there is no such concept of a 'safe set'
>
> In lines 322-355 (section 4.3) we highlight that an explicit computation of the safe sets is generally intractable, and provide a conservative approximation that is readily solvable by CMDP solvers in Eq 8. This is part of the reason that makes ActSafe scalable to deep RL settings as opposed to previous works. Moreover, prior works such as SafeOpt and GoSafeOpt, maintain a safe set of policy parameters. This restricts their application to parametric policies with only a handful of parameters. In our setting, we consider a very general class of policies often modeled through deep NNs that have millions of parameters.
> We highlight this in the practical modifications section (Section 4.3). In addition, we also mention that this relaxation maintains the safety guarantees.
>
> > In Figure 5, the author compares ActSafe with existing methods like OPTIMISTIC、UNIFORM and GREEDY. Are there any other more competitive baselines? Comparison with methods like GoSafeOpt、OPAX might be necessary.
>
> As discussed in the point above, evaluating GoSafeOpt for our setting is intractable as it is restricted to parametric policies with only a handful of parameters (generally in the order of 5-10). Furthermore, OPAX is an algorithm for unsafe exploration. We evaluate OPAX and illustrate this in our GP experiments in Figure 2 and in the Humanoid experiments in appendix C, where we observe that while OPAX achieves good performance, it leads to significant cost violations during learning. Leading us to the conclusion that it is not a competitive baseline for our experiments. However, according to the reviewer's feedback, we have added an additional experiment to our appendix, comparing OPAX and ActSafe. We note that, as expected, OPAX is capable of solving the SparseCartpoleSwingup task, however without satisfying the constraints.

---

> > ### Author Response · Authors · 2024-11-15
> > **Response to Reviewer XLTg**
> >
> > > Considering that there are no hard constraints on safety during learning in practical implementations, can ActSafe maintain safety during training after removing the initial offline data?
> >
> > Thank you for pointing this out. Generally speaking, without any prior knowledge, guaranteeing safety is  considered as an ill-posed problem. Formally, to overcome this issue, we introduce Assumption 4.3.
> > We highlight that in practice such prior can be acquired not only from offline data, but from simulators too, as we demonstrate in appendix C.
> >
> > > I am very familiar with BSRP-Lag, but the algorithm's performance in the PUSHBOX task doesn't quite match my experience, can you disclose more experimental details?
> >
> > We used a fork of the authors’ implementation, available here: https://anonymous.4open.science/r/SafeDreamer-3B72/README.md.
> > A full changelog can be found in upstream_compare.diff. We highlight that SafeDreamer’s main focus is to bound the cumulative costs to zero (i.e., ensuring state-wise safety). On its own, this is a very important challenge in safe RL. On the other hand, ActSafe focuses on ensuring constraint satisfaction during learning, with arbitrary bounds of cumulative cost. We are happy to make the above distinction more explicit if you think that would improve the paper.
> >
> > Having addressed the concerns you raised,  we kindly ask you to consider revising our score. For any remaining questions, we are happy to provide further clarification.

---

> ### Author Response · Authors · 2024-11-20
> **Followup on Review XLTg**
>
> We hope we have addressed your concerns. Let us the know if there any further questions that we can clarify, otherwise we would appreciate if you'd increase your score.
> Thanks you!

---

> ### Comment · Reviewer_XLTg · 2024-11-25
> **Response to rebuttal**
>
> I thank the authors for their efforts in addressing the rebuttal.
> - I reviewed the authors' comparative experiments incorporating OPAX into the SparseCartpoleSwingup scenarios. The experimental results aligned with my expectations, demonstrating that ActSafe's superior performance in sparse reward scenarios primarily stems from OPAX's active exploration capabilities. However, the authors' experimental section does not adequately reflect this finding.
> - To provide the authors with a clearer understanding of my concerns regarding offline data, I would like to highlight a few issues. It is evident that the baseline algorithm meets the safety constraints after several rounds of training. I suspect that the authors may have used offline data to exclude training rounds prior to the satisfaction of the safety constraint. Therefore, I would like to know whether all baseline algorithms can consistently satisfy the safety constraints after pre-training with offline data. A similar question was raised by reviewer VK9v: **Also, are all baseline algorithms utilizing this offline dataset consistently?**  I would appreciate clarification from the authors on this matter.
>
> - Additionally, my understanding is that a safe set maintains a collection of actions that satisfy the safety constraints, which is then expanded through an explicit process. This approach ensures that the safety constraints are consistently met during the training process. However, in the practical implementation, the authors transformed this into a constrained optimization problem and employed LBSGD to solve it. I am concerned that this method may not guarantee the absolute satisfaction of the safety constraints throughout the process.
>
> Since I still have some concerns about this paper, I won't raise my score for now.

---

> ### Author Response · Authors · 2024-11-25
> **Response to Reviewer XLTg**
>
> > I reviewed the authors' comparative experiments incorporating OPAX into the SparseCartpoleSwingup scenarios. The experimental results aligned with my expectations, demonstrating that ActSafe's superior performance in sparse reward scenarios primarily stems from OPAX's active exploration capabilities. However, the authors' experimental section does not adequately reflect this finding.
>
> We agree that active exploration plays a crucial role in solving sparse reward tasks. This is one of the reasons why ActSafe uses an explore—then—exploit strategy and outperforms the greedy baselines. Moreover, other methods such as SafeOpt and GoSafeOpt for safe learning use a similar approach. We highlight that while both solve the task (and other methods fail), OPAX fails to satisfy the constraints, as reflected in our experiments. This underscores the importance of intrinsic rewards in tasks that require safety, as safety introduces additional (theoretical and practical) challenges that are not accounted for in OPAX. To the best of our knowledge, this is the first paper to tackle such tasks (e.g., the sparse navigation tasks in Figure 5). What kind of change would emphasize the above in the experimental section? We are happy to accommodate your suggestions!
>
> > I would like to know whether all baseline algorithms can consistently satisfy the safety constraints after pre-training with offline data.
>
> In all our experiments, we use an initial data collection (warm-up) period of 200K environment steps, where the agent collects data and uses it to calibrate its world model. This experimental setup is simple as it seamlessly integrates with both off and on-policy algorithms, such as CPO. Furthermore, it simulates a realistic setting, where the agent can collect some data initially in a controlled/supervised setting where safety is not directly penalized. However, after the initial data collection period, the agent is required to be safe during learning. We use the same training procedure across all baselines and environments.
> Following the reviewer’s feedback, we have updated the text in Section 5.2 with this explanation and we also updated all plots to include the costs incurred during this initial data-collection phase. Furthermore, in Figure 10, we also present another experiment, where we leverage a simulation to calibrate the model and then safely fine-tune the learned model under domain shifts, effectively simulating the sim-to-real gap.
>
>
> > my understanding is that a safe set maintains a collection of actions that satisfy the safety constraints, which is then expanded through an explicit process. This approach ensures that the safety constraints are consistently met during the training process.
>
> To be more precise: the safe set maintains a collection of safe _policies_ (and not actions). Let us know if this was not clear, we are open for clarifying this further in our paper.
>
> > However, in the practical implementation, the authors transformed this into a constrained optimization problem and employed LBSGD to solve it. I am concerned that this method may not guarantee the absolute satisfaction of the safety constraints throughout the process.
>
> Indeed, your are correct that we transform the problem into a constrained optimization which we can run for the high dimensional settings considered in this work (as detailed in Section 4.3). This is because maintaining a safe set of policies in this setting is generally intractable. We highlight that this approximation still maintains all safety guarantees (see lines 346-347 in Section 4.3). As described, in our practical implementation, we use LBSGD to solve the constrained optimization problem (Equation 7). LBSGD is an interior-point method and therefore in principle all iterates during optimization are feasible. While LBSGD’s formal feasibility guarantees are restricted to a simpler setting than the deep RL setting considered in this work (see Usmanova et al. 2024), we empirically validate that it is a crucial component in reducing constraint violation compared to the augmented Lagrangian (Figure 9). This observation is in line with other works, e.g., Usmanova et al. (2024). It is important to note that our theoretical algorithm and analysis are independent to the choice of optimizer, and therefore looking forward, practitioners may use stronger solvers for constrained optimization problems.
>
> We want to thank you again for actively participating in the discussion!
> We hope this addresses your concerns and are very happy to incorporate your suggestions to improve our paper.
> We would appreciate it if you would consider a reevaluation of your score.
>
> _References_:
>
> * Ilnura Usmanova, Yarden As, Maryam Kamgarpour, and Andreas Krause. Log barriers for safe black-box optimization with application to safe reinforcement learning. JMLR, 2024.

---

> ### Author Response · Authors · 2024-11-27
> **Follow up on response to reviewer XLTg**
>
> Dear Reviewer,
>
> The 27th of November is the last day for us to make changes to the PDF. Following your suggestions, we have already updated the PDF (see our response above). We would like your feedback on our response above. If your concerns are addressed, we would appreciate it if you would consider increasing our score.

---

> > ### Author Response · Authors · 2024-12-02
> >
> > Dear Reviewer,
> >
> > We noticed that you have increased our score to a 6. We are glad we could address your concerns and thank you for increasing our score and for your active engagement during the rebuttal.

---

### Official Review · Reviewer_JgEe · 2024-11-03

**Soundness:** 3
**Presentation:** 3
**Contribution:** 2
**Rating:** 5
**Confidence:** 4

**Summary:**

This paper tackles the problem of safe exploration in reinforcement learning by proposing a new model-based constrained RL algorithm. Under the constrained MDP formulation, with regularity assumptions on the underlying dynamical system, the paper demonstrates theoretical safety guarantees and sample complexity guarantees. A practical version of the algorithm with weaker guarantees is shown to be applicable for visual control tasks in different simulated benchmarks. The results show less safety violations compared to prior constrained RL approaches.

**Strengths:**

- The paper targets a problem that is relevant to a broad community including controls, reinforcement learning, and robotics. Safe exploration in the context of model-based control is interesting because a reliable approach in this space has the potential for being deployed in safety critical scenarios as well as in scenarios that require sample efficiency and real-world exploration is challenging.

- The proposed algorithm based on intrinsic exploration for reducing uncertainty of policies at the boundary of the current safe set and expanding the safe set based on reachability beyond it, is novel to the best of my knowledge. It is also intuitively sound and doesn't make any fundamentally limiting assumptions of the underlying system.

- The theoretical guarantees of the base algorithm are strong, and the regularity assumptions seem reasonable to me.

- The experiments on simulated environments are good, and it is nice to see results in challenging control tasks like a simulated humanoid walking, that go beyond the predominant evaluations of prior safe RL works on simple simulations like SafetyGym.

**Weaknesses:**

- A major weakness is that the paper's motivations are disconnected from the experiments. For example the intro states:

"In many real-world settings, environments are complex and rarely align exactly with the assumptions
made in simulators. Learning directly in the real world allows RL systems to close the sim-to-real
gap and continuously adapt to evolving environments and distribution shifts. However, to unlock
these advantages, RL algorithms must be sample-efficient and ensure safety throughout the learning
process to avoid costly failures or risks in high-stakes applications."

However the experiments are all in simulated environments and there are no real-world experiments either in a controls setting or in a robotics setting.


- It is unclear how the model-based version of ActSafe can be reliably be deployed in the real-world for safety-critical applications. Both theoretically and empirically in the simulated experiments, the constraint violations are lower than the baselines but nowhere close to 0. In addition learning a recurrent state-space model is a more complex choice than directly trying to learn a policy through model-free RL. It is unclear why there are no evaluations in a model-free setting, and what are the motivations for just considering model-based RL, after proposing a safe exploration algorithm that seems to be fairly generic and broadly applicable. It will be helpful to clarify these points.

- In lines 233-235, it is mentioned that prior works do not have dimensional policies, but there is no explicit clarification about how high dimensional control tasks can the proposed approach tackle.  What are the relative differences? Are there any fundamental assumptions that prevent these prior works from being applicable to higher dimensional tasks?


- The related works seem to miss a lot of prior approaches that also have similar ideas of conservative exploration  and actually demonstrate results on real-world settings where safety in important. For example, see [A-C] below. [A] in particular demonstrates results on real robotic manipulation tasks where safety is critical.

[A] Thananjeyan, Brijen, Ashwin Balakrishna, Suraj Nair, Michael Luo, Krishnan Srinivasan, Minho Hwang, Joseph E. Gonzalez, Julian Ibarz, Chelsea Finn, and Ken Goldberg. "Recovery rl: Safe reinforcement learning with learned recovery zones." IEEE Robotics and Automation Letters (RA-L)

[B] Bharadhwaj, Homanga, Aviral Kumar, Nicholas Rhinehart, Sergey Levine, Florian Shkurti, and Animesh Garg. "Conservative safety critics for exploration." ICLR 2021

[C] Srinivasan, Krishnan, Benjamin Eysenbach, Sehoon Ha, Jie Tan, and Chelsea Finn. "Learning to be safe: Deep rl with a safety critic."

**Questions:**

Please refer to the list of weakness above,

- Can the authors explain the disconnect between the motivations in the paper and the experiments?

- In lines 233-235, it is mentioned that prior works do not have dimensional policies, but there is no explicit clarification about how high dimensional control tasks can the proposed approach tackle.  What are the relative differences? Are there any fundamental assumptions that prevent these prior works from being applicable to higher dimensional tasks?

- It is unclear why there are no evaluations in a model-free setting, and what are the motivations for just considering model-based RL, after proposing a safe exploration algorithm that seems to be fairly generic and broadly applicable. It will be helpful to clarify these points.

- In lines 225-226 it seems that in the general case the epistemic uncertainty will be uncalibrated / unreliable due to function approximation errors in the neural network model. Does the theory account for this?

- Can the authors clarify relations to the missing related works, including very related safe exploration works that actually show results in real-world settings?

---

> ### Author Response · Authors · 2024-11-15
> **Response to Reviewer JgEe**
>
> Thank you for taking the time to review our paper. We appreciate your acknowledgement of the importance of the problem, as well as our theoretical and empirical contributions. Please find our detailed responses to the concerns raised. Should you have any further questions or if this did not address your concerns, please feel free to reach out again.
>
> > the experiments are all in simulated environments and there are no real-world experiments either in a controls setting or in a robotics setting.
>
> We agree that deploying our work on HW will be very promising. However, we focus the scope of this paper on the algorithmic contributions. Furthermore, as is typically the case for deep RL algorithms, we evaluate them on very challenging safe RL benchmarks as also highlighted by the reviewer.
> Our evaluation in simulated environments allows us to make a comprehensive empirical analysis (as positively mentioned by all other reviewers) in a controlled experimental setup, and compare with several baselines. In appendix C.4 we provide additional experiments that demonstrate ActSafe on the Unitree H1 humanoid robot, which is known to be used in real-world sim-to-real pipelines. Nonetheless, as future work, we aim to test ActSafe directly on HW.
>
> > Both theoretically and empirically in the simulated experiments, the constraint violations are lower than the baselines but nowhere close to 0.
>
> Thank you for mentioning this point. Our problem formulation allows the designer/practitioner to choose their budget for cost return based on the problem they aim to solve, including 0. We want to highlight that the meaning of the above is that as long as the expected cumulative cost is below the specified budget, the agent is considered safe by this formulation, which is the standard approach when considering CMDPs.
> In our experiments, we follow existing evaluation protocols (e.g. Safety Gym and RWRL) to make a fair comparison.
>
> > It is unclear why there are no evaluations in a model-free setting, and what are the motivations for just considering model-based RL, after proposing a safe exploration algorithm that seems to be fairly generic and broadly applicable. It will be helpful to clarify these points.
>
> We compare ActSafe with CPO, a model-free, policy gradients CMDP solver. ActSafe relies on learning a model of the dynamics and therefore is a model-based algorithm. In figure 4 we provide a comparison of the methods within a training budget of 5000 episodes, which demonstrates ActSafe’s superior sample efficiency.
> Following your suggestion, we further highlight in lines 154-160 our motivation for using model-based methods for safety.
>
> > In lines 233-235, it is mentioned that prior works do not have dimensional policies, but there is no explicit clarification about how high dimensional control tasks can the proposed approach tackle. What are the relative differences? Are there any fundamental assumptions that prevent these prior works from being applicable to higher dimensional tasks?
>
> Thank you for pointing this out. Following your question, we highlight this further in the appendix, specifying explicitly the number of policy parameters used in our experiments (0.75M). We highlight here again, that the previous methods significantly suffer from the curse of dimensionality, rendering them impractical for problems with vision control.
>
> > The related works seem to miss a lot of prior approaches that also have similar ideas of conservative exploration and actually demonstrate results on real-world settings where safety in important.
>
> Thank you for bringing these works to our attention! Based on your suggestion, we include them in the related works section.
> We highlight that while these methods demonstrate impressive empirical results, ActSafe is also theoretically grounded.
>
> > The proposed algorithm based on intrinsic exploration for reducing uncertainty of policies at the boundary of the current safe set and expanding the safe set based on reachability beyond it, is novel to the best of my knowledge. It is also intuitively sound and doesn't make any fundamentally limiting assumptions of the underlying system.
> The theoretical guarantees of the base algorithm are strong, and the regularity assumptions seem reasonable to me.
> The experiments on simulated environments are good, and it is nice to see results in challenging control tasks like a simulated humanoid walking, that go beyond the predominant evaluations of prior safe RL works on simple simulations like SafetyGym.
>
> Thank you for acknowledging the above!
>
> We hope our responses address your concerns and would appreciate it if you’d consider reevaluating your score. We are happy to answer any further questions. Looking forward to your response.

---

> ### Author Response · Authors · 2024-11-20
> **Followup on Review JgEe**
>
> We hope we have addressed your concerns. Let us the know if there any further questions that we can clarify, otherwise we would appreciate if you'd increase your score.
> Thanks you!

---

> > ### Comment · Reviewer_JgEe · 2024-11-24
> > **Response to rebuttal**
> >
> > Dear authors,
> >
> > Thanks for writing the rebuttal response to comments. Unfortunately, I am not convinced by most of the responses and the paper edits. For example, the authors now cite related works as "These work do not ground their methods on theory" however a quick look at papers [B] and [C] shows that they indeed provide some theoretical grounding - the safety constraint violations are not zero (e.g. in A, B, C) but even the ActSafe approach doesn't provide close to 0 failures - so there is even a disconnect between theory and practice in ActSafe. More broadly, the paper's description is still disconnected to what is actually implemented. The introduction makes much grander claims than what the experiments show wrt safety in important applications, and the lack of real-world results is another limiting factor in the paper. I also see reviewer XLTg's point about disconnect b/w theory and the implementation.
> >
> > Considering all these factors, I am not about to support accepting the paper and thus will not recommend accept. I would ask the authors to consider revising the paper based on the feedback from reviewers, make the claims more consistent with the experiments, and demonstrate practical applicability of the approach on real systems (since no matter how much theoretically grounded the approach is, safety is in the end a very practical consideration)

---

> ### Author Response · Authors · 2024-11-24
> **Response to reviewer JgEe**
>
> We thank the reviewer for their response. We kindly disagree with the reviewer's assessment due to the points discussed below.
>
> **Theory for ActSafe**: We would like to emphasize that theoretical results for ActSafe give guarantees for **safety during learning** and **sample complexity** of the algorithm. To the best of our knowledge, we are the first to provide these guarantees in the context of model-based RL, and the papers mentioned by the reviewer, do not have these theoretical results. We believe that these results are important to the community since they show that ActSafe is safe for all iterations during learning while being consistent. We validate this empirically as discussed below.
>
> **Empirical evaluation**:
> 1. Our theoretical algorithm is for well-calibrated models, in particular GP dynamics. To this end, we evaluate Actsafe in Figures 2 and 3 for the GP setting and show that **ActSafe achieves the highest performance while having zero constraint violation during learning**.
>
> 2. We extend our algorithm beyond the GP setting by proposing practical modifications in Section 4.3, effectively building up on SOTA tools from deep model-based RL. Note that all implementation details of the deep RL Actsafe version are listed in Section 4.3 together with how the modifications affect our theoretical results (also discussed in our response to reviewer XLTg). We evaluate our algorithm on well-established deep RL benchmarks for CMDPs. We would like to re-emphasize that **safety in CMDPs is modeled as a constraint on the expected sum of costs** (see Equation (2)). In our experiments, we show that **ActSafe lies below the constraint thresholds** (horizontal black dashed lines in Figures 4-6 in the main paper). Therefore, we are unsure what the reviewer means by ``ActSafe approach doesn't provide close to 0 failures''. We would appreciate further clarification on this.
>
> **Experiments on real-world systems**: We believe that hardware experiments are a promising future direction. However, we are surprised that this is a criteria for accepting a methodology RL paper at ICLR.  **We evaluate ActSafe on well-established safe RL benchmarks, similar to prior works** [1--10]. Moreover, a good majority of the papers [6, 11--15], including [3, 16] are all evaluated in simulated benchmarks. We hope that the reviewer will take that into account during their evaluation.
>
> In summary:
>
> 1) We provide first-of-its-kind theoretical results for safe model-based RL methods.
>
> 2) Evaluate Actsafe in the GP setting where we empirically show safety across all learning iterations.
>
> 3) Propose and discuss practical modifications to extend Actsafe for the deep RL setting in the paper.
>
> 4) Evaluate Actsafe on well-established deep RL benchmarks for safety showing that Actsafe lies below the constraint threshold during learning.
>
>
> In light of the arguments mentioned above, we kindly ask the reviewer to reevaluate their score. We are happy to answer any further questions. Looking forward to your response.

---

> > ### Author Response · Authors · 2024-11-24
> > **Follow up on response to reviewer JgEe**
> >
> > **References**:
> >
> > 1. Weidong Huang, Jiaming Ji, Chunhe Xia, Borong Zhang, and Yaodong Yang. Safedreamer: Safe reinforcement learning with world models. ICLR, 2024.
> >
> > 2. Yarden As, Ilnura Usmanova, Sebastian Curi, and Andreas Krause. Constrained policy optimization via Bayesian world models. ICLR, 2022.
> >
> > 3. Bharadhwaj, Homanga, Aviral Kumar, Nicholas Rhinehart, Sergey Levine, Florian Shkurti, and Animesh Garg. Conservative safety critics for exploration. ICLR 2021
> >
> > 4. Alex Ray, Joshua Achiam, and Dario Amodei. Benchmarking safe exploration in deep reinforcement learning. arXiv:1910.01708, 2019.
> >
> > 5. James Queeney, , and Mouhacine Benosman. Risk-Averse Model Uncertainty for Distributionally Robust Safe Reinforcement Learning. Neurips 2023.
> >
> > 6. Aivar Sootla, Alexander I Cowen-Rivers, Taher Jafferjee, Ziyan Wang, David H Mguni, Jun Wang, and Haitham Ammar. Saute rl: Almost surely safe reinforcement learning using state augmentation. ICML, 2022.
> >
> > 7. Zuxin Liu, Zhepeng Cen, Vladislav Isenbaev, Wei Liu, Steven Wu, Bo Li, and Ding Zhao. Constrained variational policy optimization for safe reinforcement learning. ICML, 2022.
> >
> > 8. Adam Stooke, Joshua Achiam, and Pieter Abbeel. Responsive safety in reinforcement learning by PID lagrangian methods. ICML, 2020.
> >
> > 9. Yao, Yihang, ZUXIN, LIU, Zhepeng, Cen, Jiacheng, Zhu, Wenhao, Yu, Tingnan, Zhang, and DING, ZHAO. Constraint-Conditioned Policy Optimization for Versatile Safe Reinforcement Learning. Neurips 2023.
> >
> > 10. Zhan, S., Wang, Y., Wu, Q., Jiao, R., Huang, C., & Zhu, Q. (2024). State-wise safe reinforcement learning with pixel observations. L4DC 2024.
> >
> > 11. Felix Berkenkamp, Matteo Turchetta, Angela Schoellig, and Andreas Krause. Safe model-based reinforcement learning with stability guarantees. NeurIPS, 2017.
> >
> > 12. Gal Dalal, Krishnamurthy Dvijotham, Matej Vecerik, Todd Hester, Cosmin Paduraru, and Yuval Tassa. Safe exploration in continuous action spaces. arXiv:1801.08757, 2018.
> >
> > 13. Yinlam Chow, Ofir Nachum, Aleksandra Faust, Edgar Duenez-Guzman, and Mohammad Ghavamzadeh. Lyapunov-based safe policy optimization for continuous control. arXiv:1901.10031, 2019.
> >
> > 14. Joshua Achiam, David Held, Aviv Tamar, and Pieter Abbeel. Constrained policy optimization. ICML, 2017.
> >
> > 15. Danijar Hafner, Jurgis Pasukonis, Jimmy Ba, and Timothy Lillicrap. Mastering diverse domains through world models. arXiv preprint arXiv:2301.04104, 2023.
> >
> > 16. Srinivasan, Krishnan, Benjamin Eysenbach, Sehoon Ha, Jie Tan, and Chelsea Finn. Learning to be safe: Deep rl with a safety critic.

---

> ### Author Response · Authors · 2024-11-27
> **Follow up on response to reviewer JgEe**
>
> Dear Reviewer,
>
> The 27th of November is the last day for us to make changes to the PDF. Following your and other reviewers' suggestions, we have updated the PDF to further explain our experimental details and also adjusted our contributions section making it more consistent with your feedback/our empirical evaluation. We would like your feedback on our response above. If your concerns are addressed, we would appreciate it if you would consider increasing our score.

---

> > ### Author Response · Authors · 2024-12-02
> >
> > Dear Reviewer,
> >
> > The deadline for the rebuttal period is the 2nd of December. We still haven't received your feedback on our response above. We would highly appreciate hearing back from you. If your concerns are addressed, we would appreciate it if you would consider increasing our score.

---

### Official Review · Reviewer_3f3n · 2024-11-03

**Soundness:** 3
**Presentation:** 3
**Contribution:** 3
**Rating:** 8
**Confidence:** 2

**Summary:**

The authors present a model-based algorithm, called ActSafe, that has strong theoretical safety guarantees.

**Strengths:**

The paper is very well written. The high level idea of the algorithm is presented in an understandable way, without sacrificing mathematical rigour. By combining existing ideas, like intrinsic motivation or safe Expansion operators, the authors seem to have created a conceptually simple but very powerful algorithm for safe RL.

Based on a strong theoretical foundation a practical implementation is provided, that crucially maintains the safety constraints. The experiments seem well designed, highlighting the clear strengths of the algorithm (its safety guarantees), while also discussing weaknesses. The authors especially discuss scalability weaknesses that may result from the choice of Gaussian Processes to approximate the system dynamics.

**Weaknesses:**

While the two-phase approach that ActSafe employs are the foundation for its safety guarantees, I would expect that this comes at a cost. A comparison of total environment steps in both loops required, wall clock or memory requirements would have been a nice addition.

**Questions:**

Safe RL is not my area of expertise, and I therefore ask the metareviewer to discount my (unfortunately brief) review accordingly.

---

> ### Author Response · Authors · 2024-11-15
> **Response to Reviewer 3f3n**
>
> Thank you for taking the time to review our paper! Please find our responses to your questions below.
>
> > While the two-phase approach that ActSafe employs are the foundation for its safety guarantees, I would expect that this comes at a cost. A comparison of total environment steps in both loops required, wall clock or memory requirements would have been a nice addition.
>
> Thank you for raising this point. We believe that this is an important issue that safe exploration methods face. Moreover, a crucial challenge for safe exploration methods is that they have to actively select policies that reduce our uncertainty over the dynamics and thus cause our knowledge of what is safe, i.e., the safe set, to expand. Traditional exploration-exploitation trade-offs that are typically used in RL fail in this instance since for expanding the safe set we have to sample policies that may be completely suboptimal but still lead to the safe set expanding and therefore several iterations down the road to a better optimum (see Figure 1). Exploration strategies such as UCB or Thompson sampling do not account for this nuance and therefore fail in this setting (see [1]). In our experiments, figures 5 and 6 show that without this strategy, the baseline methods completely fail. Moreover, safe exploration algorithms such as SafeOpt and GoSafeOpt effectively also require this initial phase where they explore solely with respect to the model uncertainty. This challenge has also been highlighted and discussed in further detail by prior work such as [2].
>
> > Based on a strong theoretical foundation a practical implementation is provided, that crucially maintains the safety constraints. The experiments seem well designed, highlighting the clear strengths of the algorithm (its safety guarantees), while also discussing weaknesses. The authors especially discuss scalability weaknesses that may result from the choice of Gaussian Processes to approximate the system dynamics.
>
> Thank you!
>
> We hope that our answer would help you gain more confidence in our results and we are happy to answer any further questions.
>
>
> *References*
> 1. Sui, Yanan, et al. "Safe exploration for optimization with Gaussian processes." International conference on machine learning. PMLR, 2015.
> 2. Hübotter, Jonas, et al. "Transductive Active Learning with Application to Safe Bayesian Optimization." ICML 2024 Workshop: Aligning Reinforcement Learning Experimentalists and Theorists.

---

### Official Review · Reviewer_VK9v · 2024-11-04

**Soundness:** 3
**Presentation:** 3
**Contribution:** 4
**Rating:** 8
**Confidence:** 3

**Summary:**

The paper introduces ACTSAFE, a model-based reinforcement learning (RL) algorithm designed for safe exploration within continuous state-action spaces. ACTSAFE utilizes epistemic uncertainty in the model as an intrinsic reward mechanism to encourage safe set expansion during exploration. The authors provide theoretical analysis that includes safety and sample-complexity guarantees, suggesting that ACTSAFE achieves near-optimal policy convergence within finite episodes. Additionally, a practical implementation of ACTSAFE is proposed, enabling the method to scale to high-dimensional tasks, including visual control. Empirical evaluations demonstrate ACTSAFE’s performance across various benchmarks, supporting both safe exploration and strong task performance.

**Strengths:**

- Theoretical Rigor: The paper offers a thorough theoretical framework for ACTSAFE, deriving safety guarantees and sample-complexity bounds for safe exploration in continuous state-action spaces—a notable contribution in the safe RL domain.
Scalability in Practical Settings: The authors extend ACTSAFE to visual control tasks, demonstrating scalability beyond low-dimensional models. This practical application is a significant step towards bridging the gap between theoretical safe RL algorithms and real-world, high-dimensional deep RL applications.

- Comprehensive Experimental Analysis: The experimental analysis is extensive, covering both theoretical settings (under the Gaussian process assumptions) and more complex visual motor control tasks in the SAFETY-GYM environment, as well as sparse reward exploration. This breadth of evaluation provides valuable insights into the algorithm's performance under varied conditions.

**Weaknesses:**

- Reliance on Assumptions: The theoretical guarantees rely on idealized assumptions (e.g., well-calibrated Gaussian processes and specific Lipschitz conditions), which may limit generalizability. However, the authors provide strong empirical evidence that ACTSAFE performs well even when these assumptions are not strictly met, somewhat mitigating this concern.

**Questions:**

- The paper mentions the use of offline-collected data comprising 200K environment steps from a random policy. Can the authors clarify how this offline dataset is incorporated into ACTSAFE? Also, are all baseline algorithms utilizing this offline dataset consistently? If not, it may introduce unfairness in the comparisons.

---

> ### Author Response · Authors · 2024-11-15
> **Response to Reviewer VK9v**
>
> Thank you for reviewing our paper! We provide below our response to your questions. Please let us know if you have any further concerns or suggestions.
>
> > Reliance on Assumptions: The theoretical guarantees rely on idealized assumptions (e.g., well-calibrated Gaussian processes and specific Lipschitz conditions), which may limit generalizability. However, the authors provide strong empirical evidence that ACTSAFE performs well even when these assumptions are not strictly met, somewhat mitigating this concern.
>
> We agree that our theoretical guarantees and analysis require strong assumptions.
> Our assumptions are similar to other works such as SafeOpt, GoSafeOpt and while they are tough to verify in the real-world, as the aforementioned works show, these algorithms do work for safe learning in the real world. Similarly, in our experiments, ActSafe still performs well even without strictly meeting all formal assumptions. Nonetheless, we think bridging this gap between theory and practice is a very important problem that requires further attention.
>
> > The paper mentions the use of offline-collected data comprising 200K environment steps from a random policy. Can the authors clarify how this offline dataset is incorporated into ACTSAFE?
>
> Thank you for your question. In all our experiments, we use an initial data collection (warm-up) period of 200K environment steps, where the agent collects data and uses it to calibrate its world model. This experimental setup is simple as it seamlessly integrates with both off and on-policy algorithms, such as CPO. After the initial data collection period, the agent is required to be safe during learning. We use the same training procedure across all baselines and environments.
> Following the reviewer’s feedback, we have updated the text in Section 5.2 with this explanation and we also updated all plots to include the costs incurred during this initial data-collection phase.
>
> > Theoretical Rigor: The paper offers a thorough theoretical framework for ACTSAFE, deriving safety guarantees and sample-complexity bounds for safe exploration in continuous state-action spaces—a notable contribution in the safe RL domain. Scalability in Practical Settings: The authors extend ACTSAFE to visual control tasks, demonstrating scalability beyond low-dimensional models. This practical application is a significant step towards bridging the gap between theoretical safe RL algorithms and real-world, high-dimensional deep RL applications.
>
> Thank you!

---

> ### Comment · Reviewer_VK9v · 2024-11-28
>
> Thanks the authors for their rebuttals. I have no further questions now and would like to maintain my score.

---

> > ### Author Response · Authors · 2024-11-28
> > **Response to Reviewer VK9v**
> >
> > We would like to thank you again for reviewing our paper and acknowledging our work!

---

### Meta-Review · Area_Chair_X9Na · 2024-12-10

**Metareview:**

This paper proposes a model-based approach for safe RL, which has an expansion stage and an exploitation stage. The overall algorithm appears novel and the theoretical analysis is solid. The empirical study also involves challenging vision-based control tasks. So I recommend accept. That being said, I do hope to see a few changes in the camera ready version. First, I agree that hardware experiments are not necessary and it's fine to use real world problems as motivation. But the authors need acknowledge that learning a good model in most real-world problems is extremely challenging. Second, Figure 4 now has only 5 seeds. It needs to have at least 10 seeds and it's better to report mean and standard errors. SE characterizes the concentration of the empirical average and does not match with median. Third, the discussion of related works should be expanded, e.g., for "these work do not ground their methods on theory", it is necessary to make it absolutely clear what kind of theories they have and what kind of theories they do not have.

**Additional Comments On Reviewer Discussion:**

Three reviewers are in support of an accept. I read the paper and the comments of the negative reviewer. I ignored some points (e.g., the paper must have hardware experiments) and believe the rest can be addressed by some minor revision.

---

### Decision · Program_Chairs · 2025-01-22

Accept (Poster)